# Depth-dependent anisotropy in the Earth's inner core linked to chemical stratification

Efim Kolesnikov [1,4] ✉, Xiang Li [1,2], Susanne C. Müller [1,2], Arno Rohrbach [1], Stephan Klemme [1], Jasper Berndt [1], Hanns-Peter Liermann [3], Carmen Sanchez-Valle [1] & Ilya Kupenko [1,2]

Seismic anisotropy in the Earth's inner core (IC), including the heterogeneous, depth-dependent anisotropy structure, is a well-documented yet poorly understood feature plausibly related to the alignment of iron alloy crystals. Here, we report the effect of silicon and carbon on the plastic deformation of hexagonal close-packed (hcp) iron using radial X-ray diffraction at pressures up to 128 GPa and temperatures up to 1100 K. Our results reveal a low compressional wave anisotropy (~2 %) in the Fe-Si-C alloy, consistent with the seismic anisotropy observed in the outer regions of the IC. These findings, together with the higher anisotropy exhibited by pure hcp-Fe, suggest that the depth-dependent elastic anisotropy of the IC may originate from chemical stratification, i.e., radial gradients in silicon and carbon concentrations, during crystallization.

Anisotropy in compressional sound wave velocities is a well-established feature of the Earth's inner core (e.g.,[1]). Basic anisotropic models displaying cylindrical symmetry show that compressional sound waves travel around 3–4% faster parallel to the Earth's rotation axis compared to their velocity in the equatorial direction[1]. Moreover, there is evidence for more complex and intricate anisotropic patterns within the inner core[2], including relatively isotropic outer layers displaying compressional wave anisotropy of around 2% or less[3,4] and a central region exhibiting high anisotropy of up to 4–6%[1,3]. The origin of the depth-dependent anisotropy within the Earth's inner core remains, however, unclear.

Possible origins for the observed elastic anisotropy include either shape-preferred orientation (SPO), where the core-forming polycrystalline material exhibits a microstructure elongated in one or two directions, or lattice-preferred orientations (LPO), where the polycrystalline material has a preferred crystallographic alignment (texture)[5]. The presence of liquid inclusions within the solid material may result in the development of SPO[6]. However, the likelihood of that is low, as the liquid inclusions are likely to be extracted by compaction, effectively halting the development of SPO[7]. LPO in the inner core is likely to result from solidification texturing or subsequent

deformation[5,8]. The deformation should occur predominantly through dislocation creep since diffusion creep is an unfavorable mechanism for deformation development at core conditions[9] and is unable to produce LPO. The driving mechanisms of the deformation may vary between thermal convection, preferential growth in the equatorial regions, or Joule heating, with the significance of the individual processes strongly dependent on the viscosity of core materials[5].

The Earth's inner core is predominantly composed of iron alloyed with a small percentage of nickel that likely crystallizes in an hcp structure[10]. In addition, up to 5–7% of light elements alloyed to Fe−Ni are required to explain the density deficit reported by seismic models (e.g.,[11]). Geochemical and cosmochemical constraints suggest silicon, carbon, oxygen, sulfur, and hydrogen as the most probable light elements, with several of them likely coexisting in the core[12]. Recent computational and experimental studies indicate that silicon and carbon alloyed together with iron can satisfy the seismological constraints both in terms of core density and isotropic sound velocities[13,14]. The alignment of hcp crystals in the inner core as a plausible origin for the observed anisotropy in the compressional wave velocities ($V_P$) has been supported by the single-crystal elastic anisotropy (6–8%) theoretically predicted for hcp-Fe at core

[1]Institute for Mineralogy, University of Münster, Münster, Germany. [2]ESRF, The European Synchrotron, 71 Avenue des Martyrs, CS40220, 38043 Grenoble, Cedex 9, France. [3]Deutsches Elektronen-Synchrotron DESY, Notkestr. 85, 22607, Hamburg, Germany. [4]Present address: Univ. Lille, CNRS, INRAE, Centrale Lille, UMR 8207 - UMET - Unité Matériaux et Transformations, F-59000 Lille, France. ✉e-mail: ekolesni@uni-muenster.de

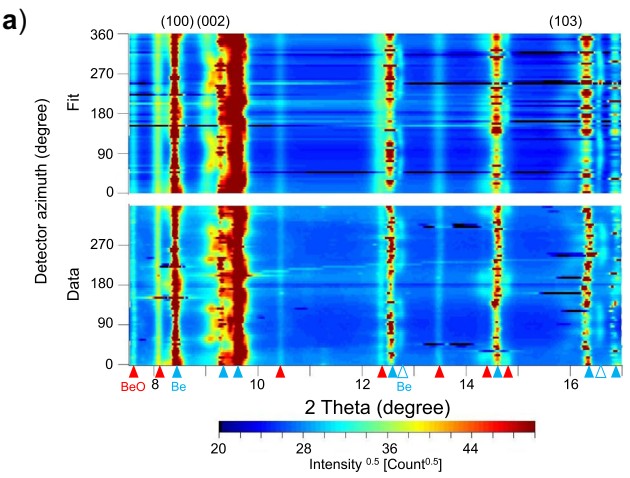

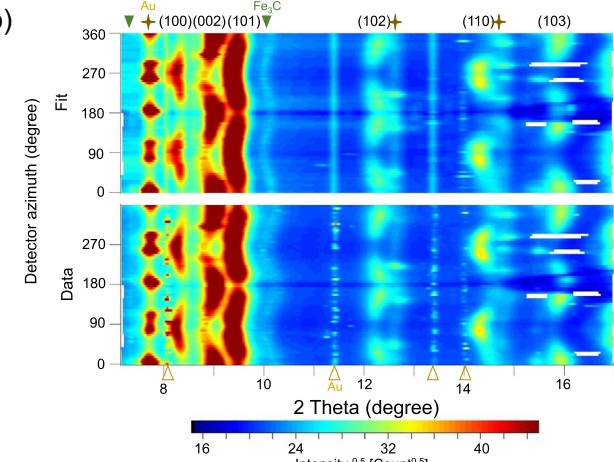

**Fig. 1 | Unrolled diffraction patterns (bottom) together with the best-fit models (top).** The intensity variations of the diffraction lines at different azimuth angles are caused by lattice-preferred orientation in hcp–Fe–2Si–0.4 C; the curvature is a measure of the elastic lattice strains. Numbers at the top (e.g., 110) indicate the Laue indices of the Bragg reflections of the compressed hcp–Fe–2Si–0.4 C sample that were used for $Q(hkl)$ calculation. **a** Pattern collected at 300 K and 107 GPa, blue-filled triangles and empty blue triangles indicate diffraction lines from the Be gasket and from Be compressed inside the sample chamber, respectively, while the red triangles show the diffraction lines of BeO formed due to oxidation of the gasket material. **b** Pattern collected at 1100 K and 100 GPa using an amorphous boron gasket, golden crosses show the diffraction lines of the Au pressure standard, the green triangles indicate $Fe_3C$ impurity, and the empty golden triangles show Au outside of the pressure chamber.

conditions[15,16] and its relatively low strength revealed by experimental studies (e.g.,[17,18]). Despite extensive research on the behavior of pure hcp-iron at high pressure regarding its strength (e.g., refs. 17,18), LPO (e.g., refs. 19–21), and sound velocity anisotropy of polycrystalline aggregate (e.g., refs. 19,22,23), the influence of alloying elements on these properties remains poorly constrained. Existing strength studies are limited to the effect of nickel[24] and silicon[25], both of which increase the yield strength of hcp-Fe[24,25], yet the combined effects of multiple light elements have not been explored. Furthermore, temperature-dependent strength measurements at high pressures remain scarce[25–27], and the impact of alloying elements on LPO and elastic anisotropy has yet to be determined.

Here, we investigate the combined effect of silicon and carbon on the deformation behavior of iron at pressures up to 128 GPa and temperatures up to 1100 K, using resistively heated diamond anvil cells (DACs) as a deformation apparatus coupled with in situ X-ray diffraction in radial geometry. We conducted deformation experiments on a polycrystalline iron-silicon-carbon alloy synthesized under high-pressure, high-temperature conditions, with a composition close to that proposed for the inner core by computational and experimental studies[13,14]: 2 wt% Si and 0.4 wt% C (hereafter referred to as Fe–2Si–0.4 C). We further employed Elasto-Visco-Plastic Self-Consistent (EVPSC) modeling to extrapolate the plastic properties of this alloy to the inner core conditions and compared results with the observed seismic characteristics of the Earth's inner core.

## Results and discussion
### Strain and strength of Fe–Si–C alloy at extreme conditions
We collected X-ray diffraction patterns of Fe–2Si–0.4 C alloy along two isotherms, at room temperature (300 K) up to 128 GPa, at 1100 ± 50 K up to 100 GPa, and along an isobar at 36 ± 3 GPa upon heating to 1100 K. Upon compression above 10 GPa, the starting body-centered cubic (bcc) structured Fe–2Si–0.4 C sample begins to transform into the hcp phase, a transformation that is complete at ~27 GPa and 300 K (Supplementary Fig. 1a, b). Upon heating, the hcp phase partially transforms into a face-centered cubic (fcc) structure at 1013 K and 39 GPa. The fcc phase is present as a less abundant component, compared to the hcp phase, and completely disappears at ~55 GPa and 1100 K (Supplementary Fig. 1c, d). We used the unrolled X-ray

diffraction patterns to determine the lattice strain parameters $Q(hkl)$, which are a measure of the amplitude of the elastic deformation of a material (see "Methods" for details). For a valid comparison of averaged lattice strain parameters $\langle Q(hkl)\rangle$ along 300 K (Supplementary Table 1) and 1100 K (Supplementary Table 2) isotherms, only the lattice strains of (100), (002) and (103) lattice planes were used for averaging, since reflections from other lattice planes are shadowed by the reflections from Be gasket for the experiment at room temperature (Fig. 1a).

When only these three lattice strains are used for averaging, the $\langle Q(hkl)\rangle$ values at 1100 K and 300 K overlap within the error bars and remain approximately 0.007 until 60 GPa (Fig. 2a). $\langle Q(hkl)\rangle$ at 300 K reaches a value of ~0.008 at 128 GPa, indicating saturation of $\langle Q(hkl)\rangle$ at the onset of compression. Conversely, at 1100 K, $\langle Q(hkl)\rangle$ starts to decrease above 60 GPa. When considering other available lattice planes at 1100 K (i.e., (100), (002), (101), (102), (110) and (103)), the averaged lattice strain parameter $\langle Q(hkl)\rangle$ is lower than presented in Fig. 2a by 8 ± 7%, although the overall trends remain the same. We observe the appearance of new reflections in the X-ray diffraction patterns that correspond to the formation of $Fe_3C$ (Fig. 1b). This observation is consistent with the results of Miozzi et al. (2022)[28] who reported the co-existence of $Fe_3C$ and hcp–Fe–Si–C alloy at high temperatures. We, thus, attribute the decrease of the $\langle Q(hkl)\rangle$ values above 60 GPa at 1100 K to the nucleation of $Fe_3C$ in the bulk Fe–2Si–0.4 C. Carbon-free compositions, like pure Fe[17], Fe–Ni alloys[24] and Fe–5 wt% Si[25], display smaller $\langle Q(hkl)\rangle$ values compared to hcp–Fe–2Si–0.4 C (Fig. 2a). Hence, the partial loss of carbon from the starting composition during heating likely causes the decrease in the averaged lattice strain parameter in the iron-silicon-carbon alloy.

The temperature effect on $\langle Q(hkl)\rangle$ is not well-constrained experimentally, and, thus, in most studies, $\frac{\partial Q}{\partial T}$ is simply assumed to be zero[17,24], providing only the upper bound for $\langle Q(hkl)\rangle$. We used our high-temperature data collected over the 36 ± 3 GPa isobar up to 1100 K (Supplementary Fig. 2, Supplementary Table 3) to better constrain $\frac{\partial Q}{\partial T}$ and obtained the value of $-8.6(\pm 2.2) \cdot 10^{-7}$ K$^{-1}$ from a set of lattice planes, namely (100), (002), (101), (102), (110), (103) and (112). Brennan et al. (2021)[25] determined $\frac{\partial Q}{\partial T}$ to be $-1.6(\pm 0.4) \cdot 10^{-6}$ K$^{-1}$ at 43–46 GPa and 300–1800 K from the $\langle Q(hkl)\rangle$ values of laser-heated Fe–Si alloys. However, their data are effectively divided into two

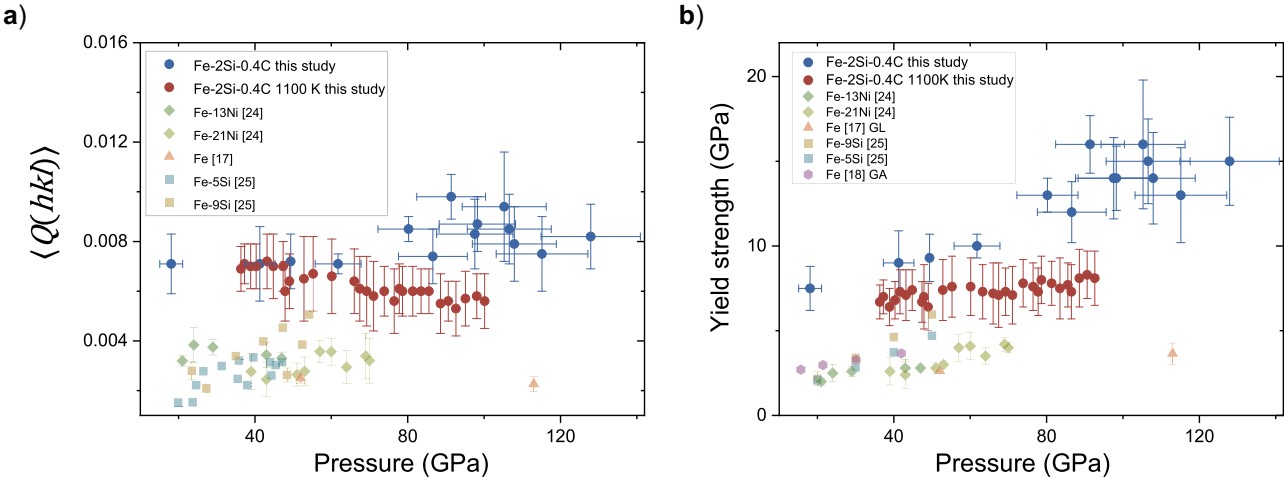

**Fig. 2 | Pressure evolution of lattice strain and strength. a** Pressure evolution of the averaged lattice strain parameter $\langle Q(hkl) \rangle$ in hcp–Fe–2Si–0.4 C at 300 K (blue circles) and 1100 K (red circles). The average is calculated from the lattice strains of (100), (002), and (103) lattice planes at both temperatures. $\langle Q(hkl) \rangle$ values of hcp–Fe–13 wt% Ni[24], hcp–Fe–21 wt% Ni[24], hcp–Fe[17], hcp–Fe–5 wt% Si[25] and hcp–Fe–9 wt% Si[25], all at 300 K, are shown for comparison. **b** Pressure evolution of

the yield strength of hcp–Fe–2Si–0.4 C at 300 K (blue circles) and 1100 K (red circles), along with literature data for hcp–Fe–13 wt% Ni[24], hcp–Fe–21 wt% Ni[24], hcp–Fe[17] (GL), hcp–Fe–5 wt% Si[25], and hcp–Fe–9 wt% Si[25], and hcp–Fe[18] (GA), all at 300 K. Error bars represent the uncertainty on pressure and $\langle Q(hkl) \rangle$ or yield strength, respectively.

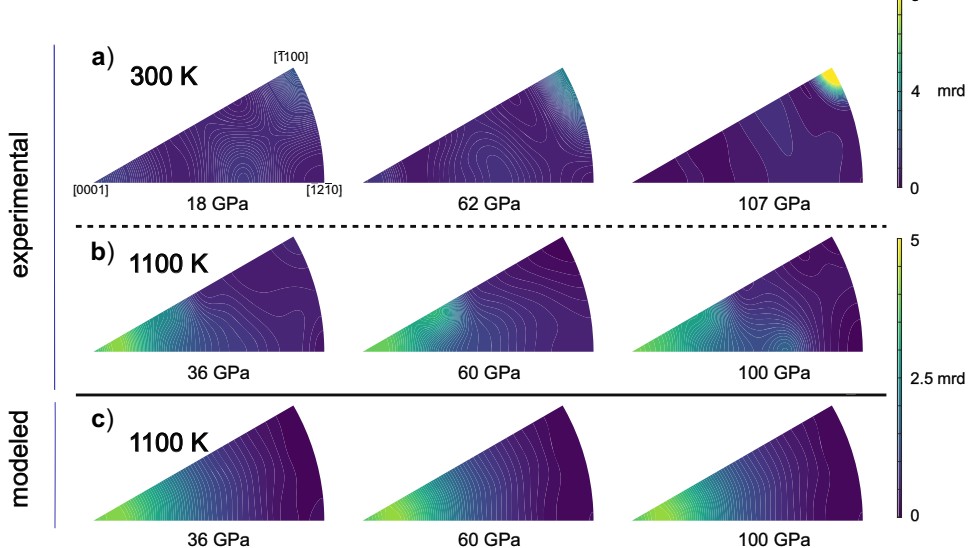

**Fig. 3 | Lattice-preferred orientation in hcp–Fe–2Si–0.4 C alloy at high pressure.** Inverse pole figures of the compression direction show experimental deformation textures in hcp–Fe–2Si–0.4 C at 300 K (**a**) and 1100 K (**b**), as well as the deformation textures resulting from EVPSC modeling at 1100 K (**c**) at the indicated

pressures. The texture intensity is expressed in multiples of random distribution (mrd), where an mrd of 1 corresponds to a totally random distribution, and an mrd of infinity corresponds to a single crystal texture.

clusters: one at 300 K and the other at 1400–1800 K (Fig. S1 in ref. 25), with a 1000 K gap. Moreover, laser heating is usually associated with large radial temperature gradients in the sample, which could explain the discrepancies in the temperature dependence of $\langle Q(hkl) \rangle$.

We applied the formalism and assumptions described in ref. 17 to derive the yield strength of the sample, which is a measure of a material's resistance to flow (see "Methods" for details). The yield strength of hcp–Fe–2Si–0.4 C at 300 K increases with pressure, reaching 15 GPa at the pressure of 128 GPa (Fig. 2b). At 1100 K, the yield strength values are lower, reaching 8 GPa at the highest pressure of 100 GPa. Across all studied pressures, the yield strength of hcp–Fe–2Si–0.4 C is higher than that of pure hcp–Fe, iron-silicon, and iron-nickel alloys (Fig. 2b). The primary compositional difference between our alloy and the Fe–Si alloys[25] is the presence of carbon in

our samples. Therefore, we conclude that the addition of carbon significantly strengthens iron-silicon alloys.

### Deformation mechanisms in iron alloys at extreme conditions

We further utilized the X-ray diffraction images to identify LPO in hcp–Fe–2Si–0.4 C (see details in "Methods"). The initial sample had extremely weak LPO at ambient conditions (Supplementary Fig. 3). Upon compression at 300 K, we observed the onset of LPO development along the [0001] direction on the inverse pole figure (IPF) in hcp–Fe–2Si–0.4 C at 18 GPa, i.e. right after the appearance of the hexagonal phase (Fig. 3a). Subsequent development of lattice-preferred orientation in hcp–Fe–2Si–0.4 C leads to a redistribution of the most intense texture direction towards $[\bar{1}100]$, reaching approximately 8 multiples of random distribution (mrd) at 107 GPa

(Fig. 3a) and further remains about the same. This redistribution can be explained by the deformation via prismatic slip, a mechanism common in other hexagonal metals[29]. At 1100 K and 36 GPa, we also observe texture along the [0001] direction. However, unlike at room temperature, [0001] remains the primary texture direction up to 100 GPa, with its intensity reaching 5 mrd (Fig. 3b). The development of [0001] textures is commonly associated with plastic deformation by basal slip $(0001)\langle\bar{1}2\bar{1}0\rangle$[20].

Further, we performed EVPSC modeling[30] that allows calculating the possible preferred orientation of a crystalline material, using its common slip and twinning systems. We compared the results of EVPSC with the experimental observations in order to identify the active deformation mechanisms responsible for the development of LPO in hcp–Fe–2Si–0.4 C at 1100 K. We only performed the modeling at 1100 K, as it is more relevant for the application of the results to the high-temperature conditions within the inner core (e.g.,[12]). For the modeling, we used stiffness tensor components at 1100 K (Supplementary Table 4, for details see "Methods"). The resulting distribution of the orientations of grains after the deformation is transformed into IPFs in the MTEX software[31]. For each deformation mechanism, we describe the hardening of the critical resolved shear stresses $\tau$ by means of an empirical linear Voce hardening rule:

$$\tau = \tau_0 + \theta_1 \Gamma \tag{1}$$

where $\Gamma$ is the accumulated plastic shear strain. The initial critical resolved shear stress (CRSS), $\tau_0$, and the asymptotic hardening rate, $\theta_1$, are the two adjustable parameters. For details, please refer to Merkel et al. (2009),[32] and references therein. We varied the activity of the various deformation mechanisms to identify the parameters (Table 1) that would allow reproducing the experimentally observed IPFs (Fig. 3b, c) and lattice strains from the full set of available lattice planes at 1100 K, i.e. (100), (002), (101), (102), (110) and (103) (Fig. 4).

We observed that, in addition to the dominant basal slip, prismatic slip, together with either pyramidal slip (Fig. 5a) or compressive twinning (Fig. 5b), is necessary to explain the observed strains (Fig. 4a) and textures (Supplementary Fig. 4).

The large scattering of experimental data prevents us from distinguishing between pyramidal slip and compressive twinning as complementary mechanisms. Furthermore, employing either of these two combinations results in the same textures, including those extrapolated to core pressures. The derived deformation mechanisms are consistent with those determined by Merkel et al. (2009, 2012)[20,32] but differ from the results of Miyagi et al. (2008)[21], who suggested dominant pyramidal slip in hcp–Fe at high temperatures. This discrepancy might arise from different pressure-temperature paths in our experiment and those of Miyagi et al. (2008),[21] or from compositional differences. Moreover, the experimental textures observed in our study for the hcp–Fe–2Si–0.4 C alloy (Fig. 3) are in line with those observed in hcp–Fe at pressures up to at least 278 GPa[33] and temperatures up to 3000 K[34]. Our experimental data, supported by the EVPSC modeling, and observations for pure iron at core conditions[35–37], allow us to assume basal slip as the main deformation mechanism in hcp–Fe–2Si–0.4 C alloy at core conditions.

## Strength and viscosity of the Earth's inner core

We employ our EVPSC deformation model to extrapolate the data and to derive $\langle Q(hkl)\rangle$ values at core pressures (Fig. 4b). Further, we

### Table 1 | List of deformation mechanisms used for the EVPSC modeling

| Model 1 | Basal | Prismatic | Pyramidal $\langle c + a \rangle$ | Compressive twinning |
|---|---|---|---|---|
| $\tau_0$, CRSS | 0.75 | 3 | 5.5 | * |
| $\theta_1$, hardening rate | 0 | 7 | 3 | * |
| Model 2 | Basal | Prismatic | Pyramidal $\langle c + a \rangle$ | Compressive twinning |
| $\tau_0$, CRSS | 0.7 | 2.6 | * | 6 |
| $\theta_1$, hardening rate | 0 | 7 | * | 3 |

Stars indicate deformation mechanisms that were not included in the particular model.
Critical resolved shear stress ($\tau_0$, CRSS) and hardening rate ($\theta_1$) are parameters for the simplified Voce hardening rule, Eq. (1), and are expressed in GPa.

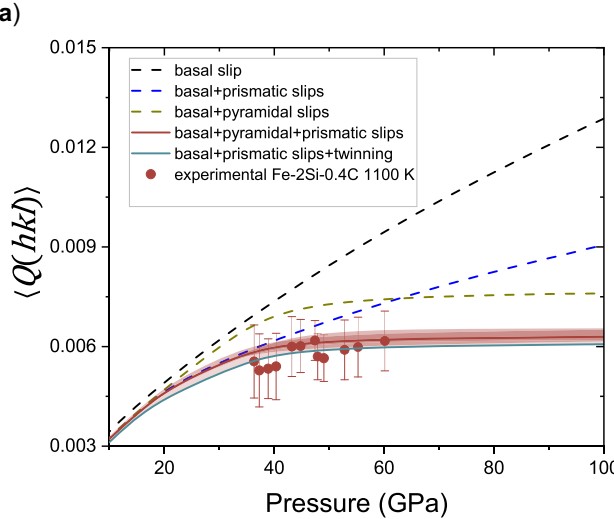

**a)**

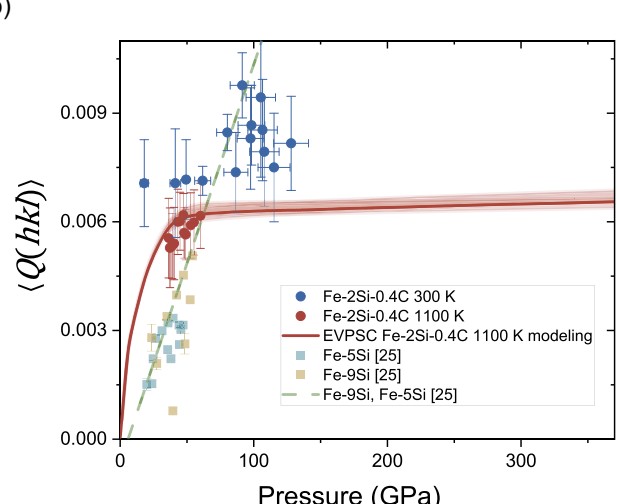

**b)**

**Fig. 4 | EVPSC modeling and experimental lattice strains. a** Experimental $\langle Q(hkl)\rangle$ values of hcp–Fe–2Si–0.4 C at 1100 K together with the results of EVPSC modeling for different active deformation mechanisms: only basal slip (black), basal + prismatic slips (blue), basal + pyramidal slips (beige), basal + pyramidal + prismatic slips (red) and basal + prismatic slips + compressive twinning (green). **b** Experimental $\langle Q(hkl)\rangle$ values of hcp–Fe–2Si–0.4 C at 300 K (blue circles) and 1100 K (red circles) together with the results of EVPSC modeling at 1100 K using

basal + pyramidal + prismatic slips (red line). Literature data for Fe–5 wt% Si (Fe–5Si, teal squares), Fe–9 wt% Si (Fe–9Si, beige squares), all at 300 K, and their linear fit (olive dash) are taken from Brennan et al. (2021)[25]. Note that here we use the full set of available lattice planes when calculating $Q(hkl)$ at 1100 K, i.e., (100), (002), (101), (102), (110) and (103) (Supplementary Table 2). Red shading represents the variation of the modeled $\langle Q(hkl)\rangle$ values and is calculated by the uncertainties propagation of the elastic constants and their pressure dependence (Supplementary Table 4).

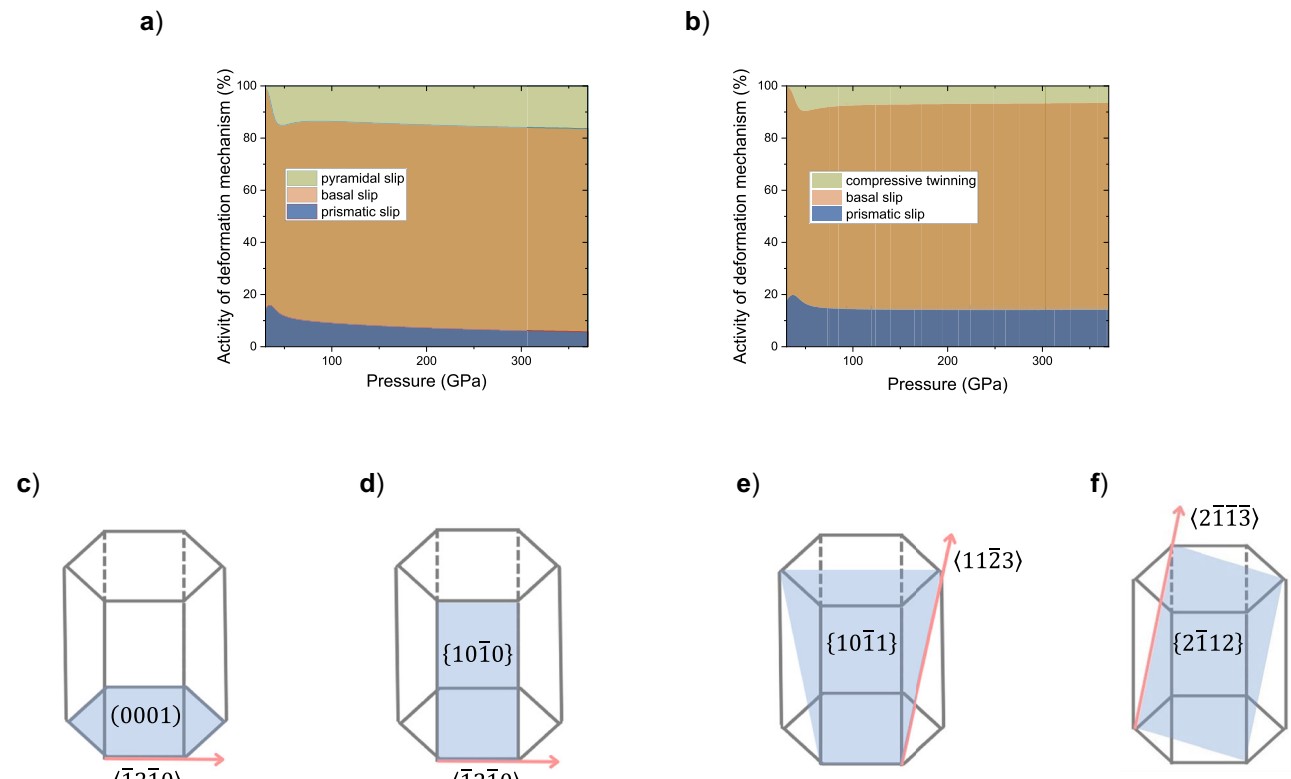

**Fig. 5 | Deformation mechanisms in hcp–Fe–2Si–0.4 C alloy at high pressure.**
Pressure dependence of the activities of various deformation mechanisms in hcp–Fe–2Si–0.4 C at 1100 K derived from the EVPSC modeling (**a**, **b**) and schematic

representation of active basal slip (**c**), prismatic slip (**d**), pyramidal slip (**e**) and compressive twinning (**f**) in hcp structure. Blue areas are slip/twinning planes, and red arrows are the corresponding slip directions.

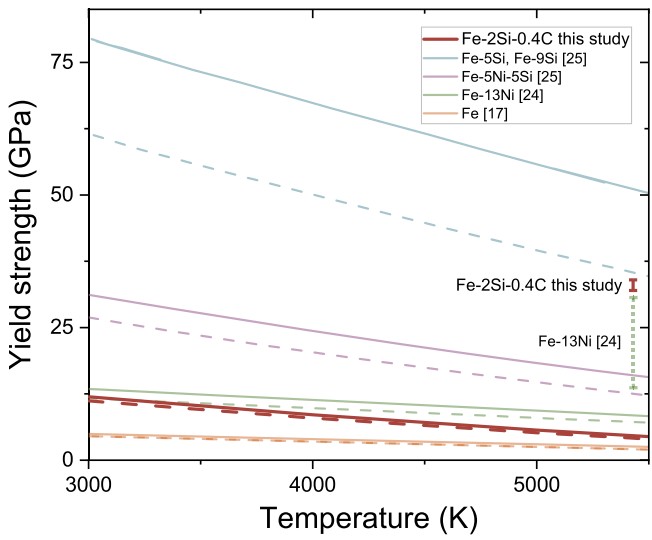

**Fig. 6 | Yield strength of hcp-Fe alloys at the core conditions.** Yield strength of different iron alloys as a function of temperature at pressures corresponding to the inner core boundary (329 GPa, dashed lines) and the center of the inner core (364 GPa, solid lines). The red error bar indicates the uncertainty in the yield strength of hcp–Fe–2Si–0.4 C at 5500 K and 364 GPa (see text for details), the dashed green error bar indicates the uncertainty in the yield strength of Fe–13 wt% Ni[24] at 5500 K and 329 GPa.

utilized the constrained $\frac{\partial Q}{\partial T}$ to calculate $\langle Q(hkl)\rangle$ at core temperatures. Finally, by combining the stiffness tensor components from the literature (Supplementary Table 5) and the extrapolated values of $\langle Q(hkl)\rangle$, we calculated the yield strength of hcp–Fe–2Si–0.4 C at

core conditions (Fig. 6, Supplementary Table 6). The uncertainty of the yield strength extrapolations was calculated from the uncertainties in the extrapolation of $\langle Q(hkl)\rangle$ to core pressures (Fig. 4b), the temperature coefficient $\frac{\partial Q}{\partial T}$ and the shear moduli (Supplementary Table 6).

Hcp–Fe–2Si–0.4 C at core conditions exhibits higher strength than pure iron (Fig. 6). The comparison with other alloys is complicated due to the inconsistencies in the available data. The strength of iron-silicon alloys as extrapolated by Brennan et al. (2021),[25] exceeds 80 GPa and is much higher than that of any iron alloys under core conditions (Fig. 6). Within the pressure range investigated in our study, however, hcp–Fe–2Si–0.4 C displays higher strength than the iron-silicon alloys studied by Brennan et al. (2021)[25] (Fig. 2b). This discrepancy between the relative magnitudes of experimentally observed and extrapolated strength values likely arises from the method of extrapolation of $\langle Q(hkl)\rangle$ values to core conditions. While we use the results of the EVPSC modeling to project $\langle Q(hkl)\rangle$ to higher pressures, which saturate due to hardening[32], Brennan et al. (2021)[25] significantly overestimated $\langle Q(hkl)\rangle$ at core conditions by linearly extrapolating low-pressure data (Fig. 4b). Similarly, the strength of Fe–Ni alloys[24] also relies on the linear extrapolation of $\langle Q(hkl)\rangle$ without considering the diminishing effect of temperature on $\langle Q(hkl)\rangle$, which leads to an overestimation of strength.

Under the assumption of deformation-induced anisotropy, the solid-state viscosity of the inner core material is the key parameter influencing the development of anisotropy[5]. We employed the values of yield strength of hcp–Fe–2Si–0.4 C predicted at the inner core conditions (Fig. 6) to calculate the sample's viscosity (see details in "Methods"). Our viscosity values (Fig. 7) for hcp–Fe–2Si–0.4 C are higher than those for pure iron derived using the same formalism[17], which is consistent with the higher strength of hcp–Fe–2Si–0.4 C (Fig. 2b). The viscosity values derived through different

approaches[9,15,38,39] show some discrepancies largely due to poorly constrained parameters under inner core conditions, albeit they generally agree on the low viscosities of pure iron, with the exception of ref. 26.

Geophysical observations indicate that the viscosity of the inner core ranges from approximately ~$10^{14}$ Pa·s (ref. 40) to ~$10^{18}$ Pa·s (refs. 41,42) (Fig. 7), which are consistent with our viscosity calculations for the hcp–Fe–2Si–0.4 C alloy. Consequently, the associated range of shear stresses in the inner core is $10^3$–$3 \cdot 10^4$ Pa, assuming hcp–Fe–2Si–0.4 C alloy is the main constituent in the inner core (Fig. 7). The estimation of the magnitudes of the shear stresses produced by various potential sources in the core is generally challenging[5]. The current estimates of stress caused by thermal convection range from $10^3$ to $10^5$ Pa (ref. 43), suggesting that thermal convection in the inner core can cause deformation-induced elastic anisotropy in hcp–Fe–2Si–0.4 C (Fig. 7).

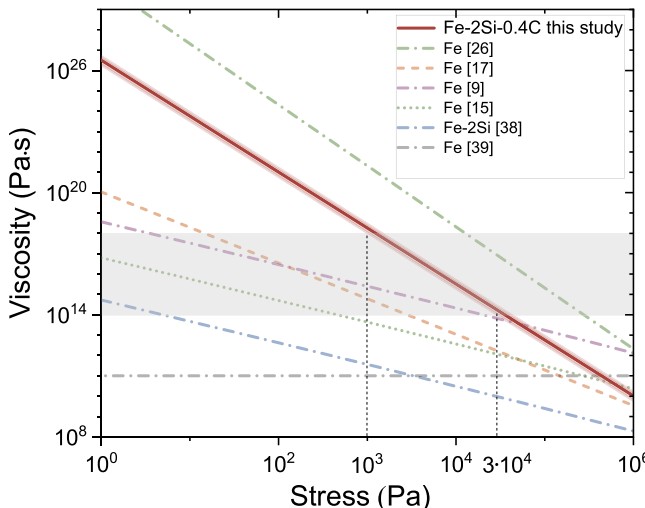

**Fig. 7 | Viscosity vs. stress at the Earth's inner core conditions.** Viscosity of hcp–Fe–2Si–0.4 C (this study) with uncertainties (red shading) and data for pure Fe[9,15,17,26,39] and Fe–2 wt% Si[38] (Fe–2Si) at 364 GPa and 5500 K. Results from ref. 39 corresponds to a temperature of 0.85-0.95 of the melting temperature. The gray-shaded area represents the range of viscosities for the inner core derived from geophysical observations[40–42,69].

## Anisotropy in the Earth's inner core

We combined experimental textures at different pressures with calculated elastic stiffness tensor components (Supplementary Table 5) to derive the anisotropy of compressional sound velocities (see "Methods" for details) and extrapolated it further to core conditions. As discussed above, basal slip is the main deformation mechanism in hcp–Fe–2Si–0.4 C both at 1100 K and at core conditions. Hence, the results of EVPSC texture modeling at 1100 K enable extrapolations of the $V_P$ anisotropy at core pressures.

The $V_P$ anisotropy of the hcp–Fe–2Si–0.4 C alloy derived from the texture at 300 K reaches 0.3-0.5% (Supplementary Table 7), which is significantly smaller than the 4-5% anisotropy observed for pure hcp-Fe at 300 K and 52–112 GPa[22,23]. Our data at 1100 K reveal the anisotropy of approximately 1% without a clear pressure trend (Supplementary Table 8), while the extrapolated anisotropy reaches about 2.0 ± 0.3% at core conditions (Fig. 8); i.e., smaller than the estimated 6.5% anisotropy of pure polycrystalline hcp-Fe at inner core conditions[19]. This indicates that Si and C significantly reduce the anisotropy of polycrystalline hcp-Fe.

The anisotropy displayed by hcp-Fe-2Si-0.4C at core conditions aligns well with the anisotropy reported by seismic studies for the outer region of the inner core which is more than 50% lower than that reported for the innermost inner core, ~4-6%[1,3,4]. Because of the higher anisotropy at core conditions of pure hcp-Fe[19] compared to hcp-Fe-2Si-0.4C alloy (Fig. 8), we hypothesize that the depth-dependent anisotropy pattern observed in the Earth's inner core may result from chemical stratification of Si and C following core crystallization. As the crystallization of the Earth's inner core begins at its center, and proceeds towards the inner core boundary across the temperature gradient[44], solidification of the core material likely starts on the iron-rich side of the eutectic through the crystallization of a hcp-Fe solid solution[12]. Whilst temperature decreases, the concentration of Si[45,46] and C[47] in the solid solution increases, suggesting a higher concentration of light elements in the direction of the inner core boundary. Concomitantly, the pressure decrease from the center to the rim of the inner core shifts the eutectic concentrations of Fe–C and Fe–Si to larger light elements concentrations[48], which should further favor the increase of Si and C concentration towards the inner core boundary. Strong chemical stratification in the Earth's inner core has been also recently proposed to explain the radial sound velocity gradients shown by PREM and AK135 seismic models[49]. While calculations of the light element gradient in the inner core requires detailed information on the phase diagram of the Fe–Si–C system at core pressures, the temperature profile of the

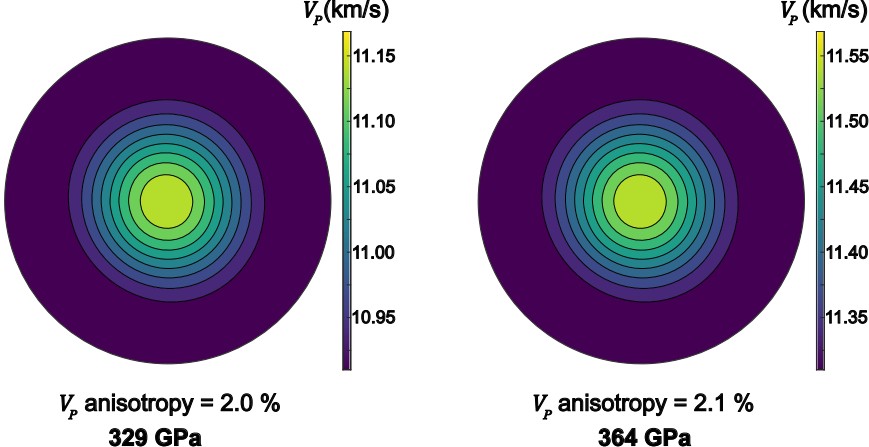

**Fig. 8 | Anisotropy of hcp–Fe–2Si–0.4 C alloy at core conditions.** Pole figure of the extrapolated compressional sound velocity ($V_P$) in a polycrystalline hcp–Fe–2Si–0.4 C aggregate along the compression direction at 5500 K and indicated pressures.

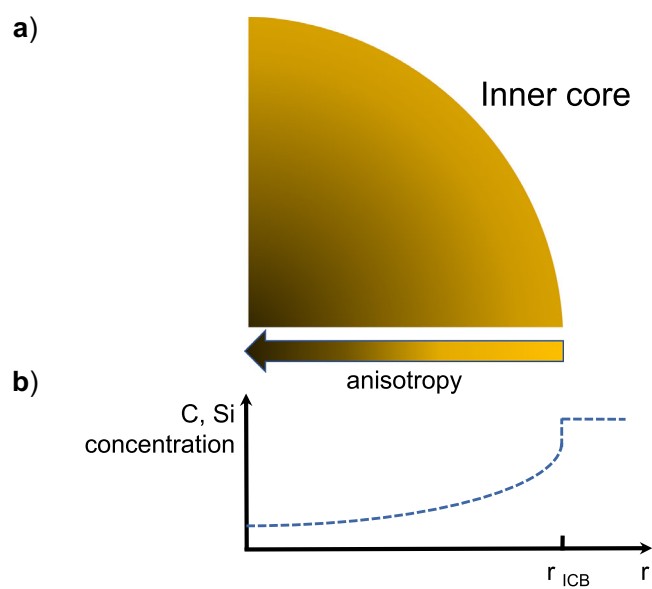

**Fig. 9 | Compressional waves anisotropy and anticipated compositional stratification in the Earth's inner core. a** Schematic representations of anisotropy variation within the inner core. **b** Qualitative distribution of Si and C concentration (following the solidus curve) within the inner core (modified after ref. 70).

outer and inner core, and a number of poorly constrained parameters(e.g.,[12]), we still can propose a qualitative trend for the variation of Si and C concentrations within the inner core (Fig. 9). The strongly anisotropic nature of the central part of the inner core can be thus linked to the depletion of Si and C in that region while the increasing concentration of light alloying elements towards the outer layers of the inner core results in reduced anisotropy.

In conclusion, we performed radial X-ray diffraction measurements on hcp-Fe-2Si-0.4 C alloy at pressures up to 128 GPa and temperatures up to 1100 K. The complementary EVPSC modeling provides robust constraints for extrapolating the combined effect of Si and C on deformation mechanisms, yield strength and viscosity of iron alloy at inner-core conditions. This represents a qualitative improvement over earlier models that relied on simple linear extrapolations and/or room-temperature data. We find that the deformation of the hcp−Fe−Si−C alloy at high-temperature, high-pressure conditions is dominated by basal slip and leads to anisotropy of compressional waves of approximately 2% at inner core conditions. This anisotropy is substantially lower than that of pure iron, and lower than seismic estimates for the center of the inner core, although it is consistent with anisotropy inferred for the uppermost inner core. These results therefore suggest that the observed heterogeneous structure of compressional-wave anisotropy within the Earth's inner core can be linked to variations in chemical composition, particularly stratification of Si and/or C.

## Methods
### Sample preparation
The Fe−2Si−0.4 C sample was synthesized from a mixture of $^{57}$Fe powder (CHEMGAS), silicon powder (Aldrich, CAS 7440-21-3) and graphite powder (Aldrich, CAS 7782-42-5). All three powder components were finely ground together in an agate mortar. The synthesis was performed in a piston-cylinder apparatus[50]. The starting material was enclosed in a BN crucible. The crucible was placed into crushable alumina cylinders (6 mm outer diameter) and surrounded by a cylindrical graphite furnace, a Duran glass cylinder (Schott, GmbH) and an outer talc sleeve. Pressure calibration was performed using the quartz-coesite transition[51] and the reaction

$MgCr_2O_4 + SiO_2 \rightarrow Cr_2O_3 + MgSiO_3$[52]. The temperature was measured by a W-Re thermocouple (Type D) and regulated to an accuracy of 5 K by a Eurotherm controller (Schneider Electric, Germany). The sample was kept at a temperature of 1673 K and 1 GPa pressure for 24 h and then quenched to below 500 degrees within <5 s by switching off the power supply. In order to avoid carbon contamination of the surface during the microprobe analyses, the sample was cleaned using a plasma cleaner and to further reduce carbon contamination in the sample chamber, a liquid-N cold trap was employed (see details in ref. 53). The composition of the product was characterized by a JEOL 8530 F electron microprobe (1−5 μm focused beam, accelerating voltage of 15 keV and beam current of 15 nA, Ir-coated sample). The matrix correction was performed relying on the φ(ρz) procedure[54]. For precise carbon calibration, a series of low-carbon steel standards (0.02 to 3.8 wt%, Micro-Analysis Consultants Ltd) and Fe$_3$C (Geller) were used to correct for the carbon background signal. The microprobe analyses showed that the sample was homogeneous in regards to the constituent elements in the sample (Supplementary Fig. 5) with the composition of the alloy being 97.63 ± 0.07 wt% Fe, 1.98 ± 0.04 wt% Si and 0.39 ± 0.03 wt% C. The sample was then removed from the microprobe mount and cut into small pieces (tens of microns in size) using a sharp razor blade and loaded in the DAC. The crystal structure of the as-synthetized sample was determined to be bcc by synchrotron powder X-ray diffraction.

### Resistive heating in diamond anvil cells
We employed DACs of the Mao-Bell piston-cylinder type to generate pressure. The sample for the room temperature experiment was compressed between two diamonds with 50 μm culets. A 20 μm hole was drilled in a Be gasket pre-indented to a thickness of about 15 μm and the sample was loaded into it. For experiments at high temperatures, a modified Mao-Bell type DAC equipped with graphite resistive heaters was employed[55]. The cell was equipped with diamond anvils with 150 μm culets. A piece of Kapton held the gaskets of amorphous boron of 25 μm thickness with ~50 μm holes in place. The entire pressure chambers were filled with the iron-silicon-carbon alloy without a pressure-transmitting medium in order to maximize non-hydrostatic stresses. For the high-temperature experiment, small pieces of gold were added to serve as a pressure standard[56]. The temperature was controlled by two type-R thermocouples, with one thermocouple in contact with the diamond close to the compression chamber and the second one in contact with the heater. A water-cooled vacuum vessel capable of maintaining $10^{-4}$ mbar was used to avoid oxidation of the diamonds, seats, and electrical connections. Additional information about the design and operation of the resistive heated DAC for X-ray diffraction in radial geometry can be found elsewhere[55].

### Synchrotron radial X-ray diffraction
The radial X-ray diffraction measurements were conducted at the Extreme Conditions Beamline of the PETRA III (DESY, Hamburg, Germany). The X-rays were oriented perpendicular to the DAC compression direction and propagated through about 3 millimeters of gasket materials (Be/amorphous boron). The X-ray beam was focused down to 8(H) × 2(V) μm² by an array of compound refractive lenses. Diffraction patterns were collected using monochromatic X-rays with a wavelength of 0.2904 Å on a Perkin Elmer XRD 1621 flat panel detector. The sample-detector distance was calibrated as 459.1 mm along with the tilt and beam center using a CeO$_2$ standard from NIST (674b) in the DIOPTAS software package[57]. We employed 3$^{rd}$ order Birch-Murnaghan equation of state (EOS) of the sample fitted to the data deposited in ref. 58 to determine the pressure in the experiment conducted at room temperature (Supplementary Table 9). The compression curve of hcp−Fe−2Si−0.4 C is close to another Fe−Si−C alloy reported before[14,59] (Supplementary Fig. 6). In the high-temperature runs, pressure was

determined using the thermal EOS of Au pressure marker[56] (Supplementary Fig. 6, Supplementary Tables 10, 11).

## Calculation of the lattice strain parameters and extraction of LPO

Diffraction images were analyzed to retrieve cell parameters and textures by the Rietveld method using the software package MAUD[60]. Diffraction images were integrated in a 5-degree step in azimuth angle which resulted in 72 slices and diffraction patterns. Representative unrolled diffraction patterns are presented in Fig. 1. The deviations of the diffraction lines from the straight line are due to the lattice strains ($\varepsilon_D(hkl, \psi)$) as described by the Eq. (2)[61]:

$$\varepsilon_D(hkl, \psi) = \frac{d_m(hkl) - d_p(hkl)}{d_p(hkl)} \qquad (2)$$

where $d_m(hkl)$ and $d_p(hkl)$ are the interplanar distances under deviatoric and hydrostatic stresses, respectively. $\psi$ is the angle between the maximum stress direction and the normal to the diffracting plane[61].

The lattice strain can be expressed as:

$$\varepsilon_D(hkl, \psi) = \left(1 - 3\cos^2\psi\right) \cdot Q(hkl) \qquad (3)$$

where coefficient $Q(hkl)$ is the lattice strain parameter for any given $hkl$ diffraction line.

Due to the weak intensity of the diffraction lines of the sample and the overlap of some diffraction lines with the diffraction lines of the gasket material, particularly at room temperature (i.e., Be, Fig. 1a), the precise derivation of the lattice strain parameters could not be performed within MAUD. Thus, we determined the peak positions of the most intense hcp–Fe–2Si–0.4 C alloy reflections in the Fityk software package and then calculated the lattice strain parameters using the formalism described above (3).

Fitting of the LPOs was performed using the E-WIMV[62] algorithm implemented with an orientation distribution function (ODF) resolution of 10° and assuming fiber symmetry about the compression direction. The E-WIMV model[62] is based on the WIMV method, extended to be used on irregular grids, which includes a built-in smoothing function. The extracted ODFs were processed using the MTEX software[31] to obtain the IPF.

## Yield strength calculations

Under the assumption of non-hydrostatic stress in a polycrystalline sample, the averaged lattice strain parameter $\langle Q(hkl) \rangle$ is linked to the uniaxial stress component $t$ and Voigt-Reuss-Hill shear modulus $G_H$ by[61]:

$$t \approx 6 G_H \langle Q(hkl) \rangle \qquad (4)$$

Uniaxial stress component $t = \sigma_3 - \sigma_1$ is a measure of the difference between axial $\sigma_3$ and radial stress $\sigma_1$ components applied to the sample. The maximum uniaxial stress $t$ sustained by a material is determined by its yield strength; that is, $t \leq \sigma_y$, where $\sigma_y$ is the material yield strength. The uniaxial stress $t$ is equal to the yield strength if the sample deforms plastically under pressure[61]. In our study, we call the uniaxial stress $t$ as yield strength for terminological consistency with other studies that used the same experimental approach (e.g.,[17,25]).

## Stiffness tensor components

Stiffness tensor components $C_{ij}$ of the hcp–Fe–2Si–0.4 C alloy at 360 GPa and various temperatures were calculated by interpolation of stiffness tensor components of Fe, $Fe_{62}C_2$, $Fe_{60}C_4$, $Fe_{60}Si_4$, $Fe_{56}Si_8$, and $Fe_{60}Si_3C_1$, and $Fe_{60}Si_2C_2$ (ref. 13), assuming independent linear compositional dependence of tensor components on Si and C content (Supplementary Fig. 7).

The pressure dependence of tensor components was derived by quadratic function $C_{ij} = aP^2 + bP + c$, where $P$ is pressure, and $a$ and $b$ are parameters dependent on Si and C content, while $c$ parameter is temperature-dependent. Parameters $a$ and $b$ were derived by fitting the pressure dependence of various iron-silicon[63] and iron-carbon[64] alloys. Parameter $c$ was adjusted from the values at different temperatures but single pressure[13] (values are in Supplementary Table 12). The derived elastic parameters of hcp–Fe–2Si–0.4 C at various pressures and temperatures employed for further calculations are summarized in Supplementary Table 4.

## Viscosity calculations

Viscosity ($\mu_{eff}$) can be defined as:

$$\mu_{eff} = \frac{\tau}{2\dot{\varepsilon}} \qquad (5)$$

where $\dot{\varepsilon}$ is a strain rate and $\tau$ is a shear stress.

The strain rate $\dot{\varepsilon}$ is defined by dislocation velocity $\nu$, Burgers vector $b$ and dislocation density $\rho_D$:

$$\dot{\varepsilon} = \rho_D b \nu \qquad (6)$$

The velocity of dislocation, $\nu(\tau, T)$, can be determined by Eq. (7)[17]:

$$\nu(\tau, T) = \frac{\nu_D a' bL}{w^2} \exp\left(\frac{-\Delta H_0}{kT}\right) \sinh\left(\frac{\Delta H_0 - \Delta H(\tau)}{kT}\right) \qquad (7)$$

where $\nu_D$ – Debye frequency, $a'$ – Peierls barrier width, $b$ – Burgers vector length, $L$ – dislocation length, $w$ – kink pair width, $\Delta H_0$ – activation enthalpy of dislocation glide at zero stress.

$\Delta H(\tau)$ changes with applied stress as[17]:

$$\Delta H(\tau) = \Delta H_0 \left(1 - \left(\frac{\tau}{\tau_P}\right)^{3/4}\right)^{4/3} \qquad (8)$$

where $\tau$ is applied shear stress, $\tau_P$ is Peierls stress which is a stress needed to move a dislocation in a crystal without thermal activation. $\tau_P$ is proportional to yield strength and we set them equal in our calculations[17].

We calculated $\Delta H_0$ as[65]:

$$\Delta H_0 = 0.317 G_H b^3 \qquad (9)$$

where $G_H$ is shear modulus, $b$ – Burgers vector.

Stress dependence of the kink pair width $w$[66]:

$$w = \sqrt{\frac{h G_H b}{8\pi\tau}} \qquad (10)$$

where $h$ is a kink pair height that is considered equal to $b$ at low stresses[65].

Density of dislocations $\rho_D$ can be expressed as[17]:

$$\rho_D = \alpha \left(\frac{\tau}{G_H b}\right)^2 \qquad (11)$$

where $\alpha$ is considered a constant of order unity[17].

Debye frequency $\nu_D$ can be described as:

$$\nu_D = \frac{\left(\frac{6\pi^2 N_a \rho}{M}\right)^{1/3} V_D}{2\pi} \qquad (12)$$

where $N_a$ is Avogadro constant, $\rho$ is the density of hcp–Fe–2Si–0.4 C calculated from the equation of state for hcp–Fe–2Si–0.4 C, assuming substitutional and interstitial incorporation mechanisms for Si and C, respectively[59], $M$ is molar mass of Fe–2Si–0.4 C iron alloy, $V_D$ is Debye sound velocity. The values of parameters used in Eqs. (6–12) are summarized in Supplementary Table 13. The density of hcp–Fe–2Si–0.4 C at the core conditions was calculated using a 3$^{rd}$ order Birch-Murnaghan EOS[58] and Mie-Grüneisen-Debye model. Parameters for the Mie-Grüneisen-Debye model calculation are taken from pure iron from refs. 67,68 The Debye sound velocity and frequency for viscosity calculations were derived from the Nuclear Inelastic Scattering (NIS) data[58]. All raw data are available in the repository associated with the manuscript (https://zenodo.org/records/15119916).

**Anisotropy calculations**

We combined experimental and modeled textures (IPFs) at corresponding pressures with calculated elastic stiffness tensor components (Supplementary Tables 4, 12, respectively) and derived the anisotropy of the compressional sound velocities in MTEX software[31]. We determine sound velocity anisotropy as:

$$Anisotropy = \frac{2 \cdot (V_{Max} - V_{Min})}{(V_{Max} + V_{Min})} \cdot 100\% \qquad (13)$$

where $V_{Max}$ and $V_{Min}$ are maximal and minimal velocities along the compression direction, respectively. The anisotropy calculation does not allow for the calculation of the uncertainty directly. However, we varied the stiffness tensor components within their uncertainty (Supplementary Tables 4, 12) to propagate the possible range of the anisotropy values of experimentally observed data (Supplementary Tables 7, 8) and of extrapolation to core conditions ($\pm 0.3\%$).

## Data availability

The diffraction data used in this study are available in the Zenodo database at https://doi.org/10.5281/zenodo.13862051.

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

## Acknowledgments

The authors acknowledge DESY (Hamburg, Germany), a member of the Helmholtz Association HGF, for the provision of beamtime. Part of the experiments were carried out at beamline P02.2 and used facilities provided by the Extreme Condition Science Infrastructure (ECSI) of PETRA III. Further thanks go to M. Trogisch for sample preparation and B. Schmitte for her support during the electron microprobe measurements. This research was partially supported by the German Research Foundation (DFG) through the DFG Project AOBJ: 662965 GZ.:KU 3832/2–1 and co-funded by the European Union (ERC, LECOR, project number 101042572). Views and opinions expressed are, however, those of the author(s) only and do not necessarily reflect those of the European Union or the European Research Council. Neither the European Union nor the granting authority can be held responsible for them.

## Author contributions

E.K., A.R., J.B. and S.K. prepared and characterized the sample. E.K., I.K., X.L., S.M., H.P.L. and C.S.-V. prepared and conducted the high-pressure, high-temperature radial X-ray diffraction experiments. E.K. analyzed the data. E.K. and I.K. interpreted the results, E.K., I.K. and C.S.-V. wrote the manuscript with contributions from all authors.

## Funding

## Competing interests

The authors declare no competing interests.
