## [Transparent Peer Review file · Nature Communications]

Depth-dependent anisotropy in the Earth's inner core linked to chemical stratification

Corresponding Author: Mr Efim Kolesnikov

Version 0:

Reviewer comments:

Reviewer #1

(Remarks to the Author)

The manuscript report diamond anvil cell experiments performed with resistive heating and radial diffraction. Data were collected along two isotherms and an isobar. The results are then modelled and extrapolated to define the main deformation mechanism in the core, used to calculate the strength, viscosity and anisotropy.

I found the manuscript interesting and praise the authors for their efforts, however I also found it hard to follow and partially incomplete. I needed multiple iterations between the manuscript, the methods and the supplementary to figure out multiple portions of the paper. And some portions are still kind of ambiguous. For example, from the displayed data it seems like the experimental data might or might not support the results of the model about the prismatic and pyramidal deformation and same for the compressive twinning, because there isn't enough resolution to discriminate. Nevertheless, the authors in the results declare an agreement with the deformation styles proposed by previous authors.

Important papers on the Fe-Si-C system relevant for the current study are not discussed (more details in the comments below). A discussion on the errors and models' uncertainties is missing, uncertainties are not reported for any of the extrapolations, and I am also wondering if the main conclusion about anisotropy in the core and the concentration curve in Figure 6 is based on thermodynamic calculations or is a representation from the authors.

By reading the current version I feel like the letter format might not be ideal for this paper, but I am happy to read an improved version once the authors have addressed my comments and questions. Until then I can't really recommend it for publication.

Major comments:

Starting material: Figure S5: The caption of the figure needs to be improved as not all the reader might have seen before chemical maps made in an EPMA. I am guessing that the interpretation of the color bar is: red pixels have a high quantity of the element, blue have a low quantity of the element? It would be useful for the authors to provide more details. And is there a reason for which the backscattered electrons image is not the same field of view of the compositional map?

- From the backscattered electron image there seems to be some preferential orientation of the grains. It is hard to tell with certainty as the polishing of the sample is not ideal (a lot of line marks). Can the authors comment on this and if possible improve on the polishing and provide a new image for the manuscript?
- Measuring carbon in the probe is challenging. Have the authors used a cold finger, N₂ and plasma cleaning of the sample surface? And can they provide a plot of the carbon content measured on the same spot at different times to show that there is no carbon contamination from the hydrocarbons likely present in the vacuum chamber of the instrument?
- How were the dimension of the measured sample and how was the sample prepared for diamond anvil cell?

- Line 288: I am a bit confused, which gasket material is giving a contribution strong enough to cover some of the peaks? From what I have read previously I thought Be and BN contribution to a diffraction pattern was minimal. Have the authors collected a diffraction pattern where the contribution of the gasket is well visible that they can show?

- The comparison with the EPSVC model is hard to follow for a reader who does not do this type of work every day. Fig.1 is clear but the informations are divided between the main text, the methods and the supplementary, which makes everything really hard to follow. Here below a series of points that I would like to see clarified:

- I see how the basal slip hypothesis is supported by the model at 1100K. Does the model also reproduce the prismatic slip as the dominant deformation at 300 K?

- Lines 93-95: If the experiments at 1100K show that the main deformation mechanism is basal slip, why do the authors say that their results are consistent with previous authors and the model where also a prismatic and pyramidal components are observed? I understand the EPSVC models show those two components should have a role too but there aren't any experiments shown having a prismatic and pyramidal contribution at high temperature. Then in the methods it is shown that the models fit the data also with compressing twin contribution and the authors mention their data are not suitable to resolve those differences, so way in the main text saying that you are consistent with previous authors? It seems to me that yes, the basal plane slip is consistent, but then there are two possible models and it's not possible to discriminate so you might or might not agree with previous authors.

- Line 332-333: Not being a specialist of deformation experiments I am failing to see how Figure 7 represent the best match between the experiments and the model. What Figure 7 is telling me (and please correct me if I am wrong) is that basal slip represents the most likely deformation mechanism? Because then when I go looking into the results the authors write that at 300K the dominant deformation mechanism observed is a prismatic slip.

- The reference for few of the key numbers used for the model and reported in Table S12 (ref 5) does not seem to be available for download. By doing a google search I have found a university of Munster page with what I guess is the description of the thesis (page only in german) but no link for download and the DOI written on the page does not produce any result when looked up online. It would be important to have this reference available to understand if a comparison of the equations of state data has been done, especially in respect to the work of Pamato et al. (2020) where the authors show that the mechanisms of solution of the carbon and silicon in the Fe produce non-negligible differences in the density of the alloy.

- Lines 134-135: If I understand correctly, we are looking at an extrapolation from 100GPa to 360 GPa. The red points representing the data at 1100K in Figure 2 do not seem to correspond to those in figure S3. In the latter the higher-pressure point for 1100K is at 75GPa while in figure 2 there are points at 100 GPa. And also, the other data seem off, has the images been rescaled in a particular manner or only certain data have been used?

- It would be useful if the authors can provide uncertainties on their numbers. Starting from the fit to obtain $\partial Q/\partial T$ and define how it propagates in pressure and temperature. Data collected at 1100K and 36 GPa are used to extrapolate values at 360 GPa and temperatures likely higher than 4000K, there need to be uncertainties.

- It would also be interesting to see how uncertainties of the parameters used for the calculation of the models affect the outcome of the investigation. Because the numbers used for the EoS are different from those reported in previous literature as for example Pamato et al. 2020 or Miozzi et al. 2020. Have the authors tried to do the same modelling using these other values as a test?

- On this, it would be useful if the authors can provide an estimate of the uncertainties also for the viscosity calculations. Many of the models/calculations come from multiple iterations of the collected experimental data plus numbers coming from other experimental and modelling work. I would like to see how the experimental error on the used parameters and the uncertainties on the previously determined parameters affect the results.

- Line 193: The reference here (N41, Komabayashi et al) discuss only the behavior of the Fe-Si system. Have the authors confronted their results with the pressure and temperature evolution of the phase diagram proposed by Hasegawa et al. 2021?

- Lines 190-194: Which data were used to calculate the evolution of the light elements concentration in the core displayed in Figure 6?

Is the increment in light elements in the solid solution related to the shift of the ternary eutectic or to the shape of the solid solution line between the solidus and the liquidus?

- Two recent papers discuss the effects of carbon and silicon incorporation in hcp Fe and the melting properties of the system respectively (Pamato et al. 2020: Equation of State of hcp Fe-C-Si Alloys and the Effect of C Incorporation Mechanism on the Density of hcp Fe Alloys at 300 K and Hasegawa et al. 2020 Liquidus Phase Relations and Solid-Liquid Partitioning in the Fe-Si-C System Under Core Pressures). A comparison of the equations and density used, along with the partitioning data used for the concentration calculations would have been appropriate, can the authors provide one? And I imagine a comparison with Pamato et al. might have been in the work cited in reference 5, but because that work seems to be not accessible it's hard to explain why the parameters of the 300K EoS from that study are different from those reported in the current manuscript.

Reviewer #2

(Remarks to the Author)

This manuscript shows X-ray diffraction measurements on an Fe-Si-C alloy at high pressure and high temperature. The sample's deformation behavior is extrapolated to Earth's core conditions. The main conclusion is that the properties of the specific composition analyzed match the seismic anisotropy in the Earth's core.

The effort the authors devoted to all the steps of the experiment and the analysis is very clear. The careful sample preparation in piston-cylinder apparatus followed by microprobe compositional analysis is very much appreciated, as well as the difficult resistive heating experiments with a panoramic diamond anvil cell. The radial X-ray diffraction data analysis is also challenging and the authors clearly show that they master all these complicated steps. However, the results are shown quickly, leaving the reader with a long list of unaddressed questions. A solid interpretation and understanding of the results is missing, while much effort is devoted to discussing extrapolated parameters in relation to Earth's core. Unfortunately, the extrapolations are presented without confidence intervals, leaving the reader uncertain about their true significance.

The structure of the manuscript is also quite unfriendly: Conclusions are missing, in the Methods are presented results (for example lines 332 – 338) and the reader continuously has to jump between methods, supplementary material, figures captions and literature to have basic fundamental information of what is being presented.

I thus believe that the authors need to revisit their work strengthening the robustness of their findings as well as improving the form of the manuscript before it can be published.

Major comments:

- Line 99: The authors state that the lattice strain parameter at 300 K and at 1100 K overlap within the uncertainty up to 60 GPa. In Line 110 the authors determine the variation of the lattice strain parameter as a function of temperature at 36 GPa. These two findings are in contradiction. From Fig. 2 the dQ/dT value seem to vary a lot with pressure (as the authors state, the 1100 K values decrease with pressure while the 300 K values increase). What is the significance of a dQ/dT calculated only at one pressure? How does this compare with the other pressures (even if with only two measured temperatures)? Moreover, the dQ/dT is compared with the literature, but the PT conditions of validity of the literature value are not mentioned.

- Line 101: The decrease is very small and within the uncertainty. Is the same decrease visible for the considered lattice planes? Or is it just an effect of the average?

The authors claim this decrease is due to the formation of cementite. Can they quantify from the diffraction patterns the amount of cementite present in their sample and check if this is consistent with their hypothesis? If cementite is forming it means that the alloy is heterogeneously depleted in carbon and more silicon rich. How would this affect the results shown?

- Fig.2 : Fe-5Si data from ref [18] are shown, while in Supplementary Fig. 3 are shown Fe-5Si and Fe-9Si data from ref [10]. If they are the same data, please refer to them consistently. Why are Fe-9Si data missing in Fig.2? Please specify in the caption the temperature of all data sets.

- Supplementary Fig.1: it would be useful to show here which lattice planes were used for the determination of Q.

Minor comments:

- Line 65: probably here it's worth mentioning that the sample is polycrystalline and synthesized at high pressure and temperature. Why did the authors choose this specific composition?

- Line 78: The scale indicating the multiples of mrd in Fig. 1 goes maximum to 4. I understand that the scale is not the same for the 300 K and the 1100 K cases.

- Line 137: Too many information missing. I suggest you to change the sentence with something like: "Finally, by combining the stiffness tensor components from the literature and the extrapolated value of Q, we calculated the yield strength of Fe-2Si-0.4C at core conditions."

The uncertainty of this value, as shown in Supplementary Fig. 4 looks very small. Have the authors propagated the uncertainties over the extrapolation? In any case, it is worth mentioning how this uncertainty was determined, before claiming that it is smaller than the previous one.

- Line 222: some "shades" are visible in the backscattered electron image, while the compositions look completely homogeneous. Are the two images taken on the exact same spot?

Can you add a compositional scale to the elemental X-ray distribution maps? What do the colors represent?

Version 1:

Reviewer comments:

Reviewer #1

(Remarks to the Author)

I have now looked at the revised version of the manuscript "Depth-dependent anisotropy in the Earth's inner core linked to chemical stratification" by Kolesnikov and co-authors. Before getting into the details, I want to apologize with the authors for the delay of my review and thank them for the thorough response to my previous comments.

The authors made a great effort in making the manuscript more accessible and clearer for the reader and I think it has improved in respect to the first version, but it still requires some work.

I have few more questions and some comments that I think will improve the paper. After this I think the work will be a nice contribution to the field.

The authors start discussing the results without mentioning what is the stable crystallographic structure and potentially the chemistry of the sample. Only after a while hcp Fe-2Si-0.4C starts being mentioned. It would be great if a mention to the structure and chemistry could be added so that a reader can understand better.

On the iron side, starting at 36 GPa with the high temperature data the authors might cross the fcc stability field before the hexagonal structure becomes the stable one? It would be great if the author can comment on this. The only mention of the chemistry comes in line 89 where the manuscript states "We observe the appearance of new reflections in the X-ray diffraction patterns that correspond to the formation of Fe₃C" and then in line 90 there is a reference about hcp being stable with Fe₃C but there are no reference to what are the structures are before that or even in the image of the diffraction patterns in the supplementary.

It should also be mentioned what is the initial structure of the alloy (maybe in the methods where the sample is described?) and not only in the caption of supplementary Figure 3.

Additionally, the Fe-Si-C alloy is defined in different ways across the paper, e.g., it starts as hcp Fe-2Si-0.4C, then in line 360 it is defined as hcp Fe alloy then in line 190 and further down it is referred to as Fe-2Si-0.4C. I'd suggest using one of

the three across all the parts of the paper to improve clarity for the reader.

On this note, I think Figure Supplementary 1 needs some clarity. This is the first image in which raw data are shown, it would be great if there can be some more details to make it more understandable. What is the color scale? And what are the other phases for the 1100 K data? Why in the figure at 1100 K only the peaks of Fe₃C are shown?

I would also add to the caption of figure b that it is to show the contribution of the gasket to the diffraction pattern, as readers might wonder why there is no sample listed.

“Carbon-free compositions, like pure Fe₁₇, Fe-Ni alloys²⁴ and Fe-5 wt % Si²⁵, display smaller $\langle Q(hkl) \rangle$ values compared to Fe-2Si-0.4C (Fig. 1a). Hence, the partial loss of carbon from the starting composition during heating likely causes the decrease in the averaged lattice strain parameter in the iron-silicon-carbon alloy.”

The data the authors are comparing with, are data collected with another scope in mind and consequently trying to be as hydrostatic as possible and with another beamline geometry (axial and not radial). Because their conclusion for this portion is based on the comparison, can the authors comment on how reliable it is to compare datasets collected in such different ways?

In line 252 the authors mention the concentration of C in the solid solution changing with temperature

The data proposed were collected in axial standard geometry, does it matter?

Line 46: “explain the density deficit compared to reported by seismic models” Seems like maybe only compared to or reported by should be kept?

Line 76 and 78: $Q(hkl)$ are written with different formalisms.

Line 81: Typo: “lattice stains”

Line 87: “Compared to presented” there is a subject missing?

Line 90: “co-existence of Fe₃C and hcp-Fe-Si-C alloy” I think Miozzi et al. observed hexagonal structured iron from the diffraction pattern and they used the volume increase to infer the presence of Si and C in the structure of the iron. Is the same structuring and volume observed for the “material” in this work to confirm that indeed is a similar phase?

Line 97: “providing only the upper boundary for $\langle Q(hkl) \rangle$.” What do the authors mean?

Supplementary figure 1: The spacing between top and bottom needs to be bigger. Both the figures look like it is only one and not two.

Line 176: “to derive $\langle Q(hkl) \rangle$ values that are further utilized the constrained $\partial Q / \partial T$ to calculate $\langle Q(hkl) \rangle$ at core temperatures.”

Not clear what the authors meant here, maybe it was “to constrain $\partial Q / \partial T$ ”?

Lines 193-196: “Within the pressure range investigated in our study, however, hcp-Fe-2Si-0.4C displays higher strength than the iron-silicon alloys studied by Brennan et al. (2021)²⁵ (Fig. 1b). This discrepancy likely arises from the extrapolation of $\langle Q(hkl) \rangle$ values to core conditions.” I am confused by this sentence, how can a discrepancy in the studied pressure range depend on an extrapolation at core condition? Maybe the authors meant something else?

Line 232: “experimental Vp anisotropy”. If the Vp is estimated (line 231) from the texture modelling I think it shouldn't be called “experimental”. It might lead people to think it was an experimentally observed quantity, while it is an estimate, from a model in turn based on the experiments.

Reviewer #2

(Remarks to the Author)

The authors indeed put some effort in trying to ameliorate the structure of their manuscript, which now reads better than the previous version. However, the manuscript still does not stand alone as too many essential information (especially Figures) have to be looked for in the supplementary materials to be able to understand what the authors are talking about.

Moreover, the authors tried to answer the question about previous line 101 (current 84) referencing to the data in table 2 of supplementary materials. I plotted these data and I performed a linear fit of the average over all the lattice planes indicated in line 86. The result I obtain is $y = 9e-06 x + 0.0053$, which is increasing over pressure and not decreasing as stated in lines 85-87. The statement in line 91 and the related conclusions are thus not supported by the data presented in the paper. The statement in line 91 is also in contradiction with what is shown in Supplementary Figure 5.

I thus feel that still some work has to be done to make this manuscript ready for publication and that Nature Communications is probably not the suitable choice of journal.

Version 2:

Reviewer comments:

Reviewer #1

(Remarks to the Author)

I have now looked at the revised version of the paper. I think the manuscript reads well in its current form and I thank the authors for their work. They clarified the points that I had questions on, and made the manuscript approachable despite the short format. I have no further comments.

Reviewer #2

(Remarks to the Author)

I apologize with the authors for my mistake trying to verify their results. Their statement about the pressure evolution of the averaged lattice strain parameter $\langle Q(hkl) \rangle$ was indeed correct.

The manuscript indeed reads much better now than in the previous versions, and the findings are more clearly presented and discussed.

Still, I don't understand the relevance of showing lattice strain parameter averaged on a different set of reflections (Fig. 2 and Fig. 4.), especially given that the trend of the average over pressure changes according to the number of reflections considered: for example, in Fig 2 (average over 3 reflections) the general trend is that $d\langle Q \rangle/dP$ is negative, while in Figure 4 (average over 6 reflections) it is positive.

Moreover, I think it would be appreciable to have a more developed and dedicated conclusion paragraph.

In conclusion, in this manuscript it is shown how the anisotropy is dependent on the chemistry. Measurements up to about 100 GPa and 1100 K are used to model the deformation mechanisms, yield strength and viscosity of hcp Fe-2Si-0.4C at core conditions. My only reservation is whether such a technical work meets the criteria for a high-impact journal.

Response to Reviewer 1:

The manuscript report diamond anvil cell experiments performed with resistive heating and radial diffraction. Data were collected along two isotherms and an isobar. The results are then modelled and extrapolated to define the main deformation mechanism in the core, used to calculate the strength, viscosity and anisotropy. I found the manuscript interesting and praise the authors for their efforts, however I also found it hard to follow and partially incomplete.

We thank Reviewer #1 for acknowledging the importance of our results and for the positive evaluation of our experimental work.

I needed multiple iterations between the manuscript, the methods and the supplementary to figure out multiple portions of the paper. And some portions are still kind of ambiguous. For example, from the displayed data it seems like the experimental data might or might not support the results of the model about the prismatic and pyramidal deformation and same for the compressive twinning, because there isn't enough resolution to discriminate. Nevertheless, the authors in the results declare an agreement with the deformation styles proposed by previous authors. Important papers on the Fe-Si-C system relevant for the current study are not discussed (more details in the comments below). A discussion on the errors and models' uncertainties is missing, uncertainties are not reported for any of the extrapolations, and I am also wondering if the main conclusion about anisotropy in the core and the concentration curve in Figure 6 is based on thermodynamic calculations or is a representation from the authors. By reading the current version I feel like the letter format might not be ideal for this paper, but I am happy to read an improved version once the authors have addressed my comments and questions. Until then I can't really recommend it for publication.

Answer:

Following the reviewer's suggestions we have reorganized the structure of the manuscript to increase its clarity. Added/modified parts are highlighted in yellow, and rearranged parts are highlighted in turquoise. The responses to specific points of criticism are provided below.

Major comments:

Question: Starting material: Figure S5: The caption of the figure needs to be improved as not all the reader might have seen before chemical maps made in an EPMA. I am guessing that the interpretation of the color bar is: red pixels have a high quantity of the element, blue have a low quantity of the element? It would be useful for the authors to provide more details.

Answer: Indeed, the interpretation of the color maps is as suggested by the reviewer. We have added this clarification to **Supplementary Fig. 6** (former **Supplementary Fig. 5**). Please note, that the color maps there are qualitative and, thus, the quantitative legend cannot be provided.

Question: And is there a reason for which the backscattered electrons image is not the same field of view of the compositional map? - From the backscattered electron image there seems to be some preferential orientation of the grains. It is hard to tell with certainty as the polishing of the sample is not ideal (a lot of line marks). Can the authors comment on this and if possible improve on the polishing and provide a new image for the manuscript?

Answer: The compositional maps were taken from the same spot as the backscattered electron image. We have taken compositional maps only in the squared form (the centers of images are

the same). We have modified the figure caption accordingly. The differences in the SEM image between lighter and darker areas are caused by electron channelling due to different crystal orientations. We have checked also the LPO in non-compressed Fe-2Si-0.4C alloy and it is extremely weak (below 1.1 mrd, see **Supplementary Fig. 3**). The line marks are indeed scratches from imperfect polishing. Unfortunately, the sample is no longer accessible since it was fully used in the experiments. Therefore, we cannot provide a new image or improve the polishing.

Question: Measuring carbon in the probe is challenging. Have the authors used a cold finger, N₂ and plasma cleaning of the sample surface? And can they provide a plot of the carbon content measured on the same spot at different times to show that there is no carbon contamination from the hydrocarbons likely present in the vacuum chamber of the instrument?

Answer: Yes, we used plasma cleaning (to avoid C contamination at the surface), and N₂ and cold finger. Details are given in Berndt & Klemme (2022) in the methods section. We have modified the text accordingly. The sample is no longer accessible, therefore we cannot provide carbon content on the same spot at different times. We used a series of low-carbon steels and Fe₃C for blank carbon correction. We have added this information to the methods section (lines 301-304, 307-309).

Question: How were the dimension of the measured sample and how was the sample prepared for diamond anvil cell?

Answer: The initial sample size was on the order of several millimetres. In order to load it to the DAC, we took the sample from the microprobe mount and then cut small pieces (tens of microns) with the razor blade to fully fill in the pressure chambers with them. We have added this information to the text (lines 312-313).

Line 288:

Question: I am a bit confused, which gasket material is giving a contribution strong enough to cover some of the peaks? From what I have read previously I thought Be and BN contribution to a diffraction pattern was minimal. Have the authors collected a diffraction pattern where the contribution of the gasket is well visible that they can show?

Answer: The material that creates additional diffraction lines is the beryllium gasket. It has Be and BeO in the composition. The experiment geometry requires the orientation of the DAC compression direction to be perpendicular to the X-ray beam, and, thus, the X-ray has to propagate through about 3 mm of the gasket material before reaching the samples. Therefore, although being made of light material, the gasket contributes a lot to the pattern compared to the sample due to several orders of magnitude higher volume exposed to the X-ray beam. We have added a diffraction pattern with the contribution of Be and BeO as **Supplementary Fig. 1b** and to the text (lines 332-334).

Question: The comparison with the EPSVC model is hard to follow for a reader who does not do this type of work every day. Fig.1 is clear but the informations are divided between the main text, the methods and the supplementary, which makes everything really hard to follow. Here below a series of points that I would like to see clarified:

I see how the basal slip hypothesis is supported by the model at 1100K. Does the model also reproduce the prismatic slip as the dominant deformation at 300 K?

Answer: We have performed EVPSC modeling for our results at 1100 K, but not at 300 K. The EVPSC modeling requires significant computational resources, while results at 300 K would be still irrelevant for the inner core. We have added this information to the text (lines 147-151). Therefore, we made assumptions based on the literature results on the prismatic slip in hcp structures (Pochettino et al. (1992)). In addition, we have made one simulation of deformation with activation of only prismatic slip in hcp and present the resulting inverse pole figure in **Response Fig. 1** here below. The direction of the LPO development in **Response Fig. 1** is the same as at 300 K upon compression to above 60 GPa (**Fig. 2a**) (former **Fig. 1a**).

Response Fig. 1. Simulation of deformation by a single prismatic slip in an hcp structure (not the modeling to fit the actual data).

Lines 93-95:

Question: If the experiments at 1100K show that the main deformation mechanism is basal slip, why do the authors say that their results are consistent previous authors and the model where also a prismatic and pyramidal components are observed? I understand the EPSVC models shows those two components should have a role too but there aren't any experiments shown having a prismatic and pyramidal contribution at high temperature. The in the methods it is shown that the models fit the data also with compressing twin contribution and the authors mention their data are not suitable to resolve those differences, so way in the main text saying that you are consistent with previous authors? It seems to me that yes, the basal plane slip is consistent, but then there are two possible models and it's not possible to discriminate so you might or might not agree with previous authors.

Answer: The deformation usually occurs by a combination of several different mechanisms. We refer to the studies of Merkel et al. (2012) and Miyagi et al. (2008) performed on hexagonal iron at high temperatures (600-1950 K). The results of Miyagi et al. (2008) indeed differ from ours, and we have changed the text accordingly. Our results are consistent with Merkel et al. (2012) in considering basal slip as the dominant deformation mechanism. However, in addition to the basal slip, Merkel et al. (2012) also suggest twinning, prismatic and pyramidal slips as complementary active deformation mechanisms at high temperatures. We have checked our statement in the text and modified it to be more precise (lines 153-163). In **Supplementary Fig. 5a** we show that a single basal slip cannot explain the development of $\langle Q(hkl) \rangle$, and the addition of prismatic and pyramidal slips or twinning is necessary.

Lines 332-333:

Question: Not being a specialist of deformation experiments I am failing to see how Figure 7 represent the best match between the experiments and the model. What Figure 7 is telling me (and please correct me if I am wrong) is that basal slip represents the most likely deformation mechanism? Because then when I go looking into the results the authors write that at 300K the dominant deformation mechanism observed is a prismatic slip.

Answer: **Fig. 3** (former **Fig.7**) does not show the likelihood of the deformation mechanism but the mechanisms active at various pressures based on our EVPSC model, that matches best to experimentally observed textures (**Fig. 2**) and lattice strains (**Supplementary Fig. 5**). **Fig. 3** indeed shows that the basal slip is the most active deformation mechanism at 1100 K. Please note, that **Fig. 3** depicts the results for 1100 K not for 300 K.

Question: The reference for few of the key numbers used for the model and reported in Table S12 (ref 5) does not seem to be available for download. By doing a google search I have found a university of Munster page with what I guess is the description of the thesis (page only in german) but no link for download and the DOI written on the page does not produce any result when looked up online. It would be important to have this reference available to understand if a comparison of the equations of state data has been done, especially in respect to the work of Pamato et al. (2020) where the authors show that the mechanisms of solution of the carbon and silicon in the Fe produce non-negligible differences in the density of the alloy.

Answer: We have modified the link for the online available data (<https://zenodo.org/records/15119916>). We have added to this the pressure-volume data, the density of phonon states and derived Debye velocities. The detailed comparison of the EOS parameters with the results of Pamato et al. (2020) would be indeed interesting, but falls out of the scope of the present study and will be published elsewhere. We add the data of Pamato et al. (2020) to **Supplementary Fig. 7**, and their results are quite close to ours, particularly at higher pressures. The existing discrepancy between our results and those of Pamato et al. (2020) likely comes from compositional differences. Note, that we assume the same incorporation mechanism for Si and C as Pamato et al (2020) and we highlighted that now in the text (lines 433-434). We are currently preparing for submission of another paper dedicated specifically to the EOS of Fe-Si-C alloys where we do a detailed comparison. Please also note, that the derived equation of state is only used in the current manuscript for calculating the pressure for the 300 K experiment (thus not affecting the data at 1100 K used for the extrapolation to core conditions) and for the viscosity calculations we address this point in the response below.

Lines 134-135:

Question: If I understand correctly, we are looking at an extrapolation from 100 GPa to 360 GPa. The red points representing the data at 1100K in Figure 2 do not seem to correspond to those in figure S3. In the latter the higher-pressure point for 1100K is at 75GPa while in figure 2 there are points at 100 GPa. And also, the other data seem off, has the images been rescaled in a particular manner or only certain data have been used?

Answer: Indeed, the red points representing data at 1100 K do not match in **Fig. 1a** (former **Fig.2**) and **Supplementary Fig. 5** (former **Fig. S3**). In **Fig. 1a** we plot data at 1100 K as averaged over (100), (002), and (103) lattice planes for a direct comparison with the data at

300 K. In **Supplementary Fig. 5** we plot the averaged lattice strain parameter at 1100 K over a more complete set of the lattice planes ((100), (002), (101), (102), (110), (103)) that were accessible in the high-temperature data set. We used the same set of lattice planes from the EVPSC modeling for comparison with the experimental data. In **Supplementary Fig. 5**, we use the pressure points without significant precipitation of cementite Fe_3C , i.e. until 60 GPa. We have modified the captions to make it clearer and added clarifications to the text (lines 77-81, 117-118). The other common data plots in **Fig. 1a** and **Supplementary Fig. 5** are Fe-2Si-0.4C at 300 K, Fe-5Si and Fe-9Si, they are plotted from the same data but may look differently due to the different scales in the two figures.

Question: It would be useful if the authors can provide uncertainties on their numbers. Starting from the fit to obtain $\partial Q/\partial T$ and define how it propagates in pressure and temperature. Data collected at 1100K and 36 GPa are used to extrapolate values at 360 GPa and temperatures likely higher than 4000K, there need to be uncertainties.

Answer: We determined the value of $\frac{\partial Q}{\partial T}$ as $-8.6(\pm 2.2) \cdot 10^{-7} \text{ K}^{-1}$ from the fit in **Supplementary Fig. 2** (also lines 99-100 in the main text). It would be indeed interesting to see the uncertainty propagation of $\frac{\partial Q}{\partial T}$ with pressure and temperature. However, its experimental determination is already quite challenging. Please note that in most of the previous studies $\frac{\partial Q}{\partial T}$ was simply assumed to be zero. The value, determined in our study, although at a single pressure point and a limited temperature range, thus, provides some experimental estimation.

Question: It would also be interesting to see how uncertainties of the parameters used for the calculation of the models affect the outcome of the investigation. Because the numbers used for the EoS are different from those reported in previous literature as for example Pamato et al. 2020 or Miozzi et al. 2020. Have the authors tried to do the same modelling using these other values as a test?

Answer: Our EVPSC modelling does not directly provide the uncertainties. However, in order to estimate the uncertainties of the model, we have varied all the stiffness tensor components within their uncertainties (**Supplementary Table 13**), thus resulting in a range of possible $\langle Q(hkl) \rangle$ values up to core conditions that are now displayed in **Supplementary Fig. 5**. Estimation of uncertainty for anisotropy was also added (line 236, lines 444-448) Parameters of the EOS are not involved in the calculation of the EVPSC model. However, we have verified the effect of the choice of the EOS, i.e. the one presented in the manuscript and of Pamato et al. (2020). in the viscosity calculations (see below).

Question: On this, it would be useful if the authors can provide an estimate of the uncertainties also for the viscosity calculations. Many of the models/calculations come from multiple iterations of the collected experimental data plus numbers coming from other experimental and modelling work. I would like to see how the experimental error on the used parameters and the uncertainties on the previously determined parameters affect the results.

Answer: We have employed the uncertainties of parameters in **Supplementary Table 14** and calculated the uncertainty of viscosity that is now added to **Fig. 5**, line 221. In particular, viscosity derived with the application of the parameters from the EOS of Pamato et al. (2020) falls within our uncertainty range.

Line 193:

Question: The reference here (N41, Komabayashi et al) discuss only the behavior of the Fe-Si system. Have the authors confronted their results with the pressure and temperature evolution of the phase diagram proposed by Hasegawa et al. 2021?

Answer: Following the reviewer's suggestion we have modified the references accordingly. We need to clarify that our statement here also comes from the general suggestion of the form of the solidus curve. The results of Hasegawa et al. (2021) also support our results. Pressure decrease shifts the eutectic concentrations of Fe-C and Fe-Si to larger light elements concentrations. Therefore, it is intact with the idea that Si and C concentration increases to the inner core boundary. We have clarified these points in lines 253-258.

Lines 190-194:

Question: Which data were used to calculate the evolution of the light elements concentration in the core displayed in Figure 6? Is the increment in light elements in the solid solution related to the shift of the ternary eutectic or to the shape of the solid solution line between the solidus and the liquidus?

Answer: The evolution of the light elements concentration in now **Fig. 7** (former **Fig. 6**) was not calculated, it is a schematic representation based on the results of Fischer et al. (2013) and Ozawa et al. (2016) for Si and Mashino et al. (2019) for C and a general understanding of the form of the solidus curve. Calculation of exact amounts of the light elements in the inner core requires detailed information on a phase diagram of the Fe-Si-C system at core pressures, the temperature profile of the outer and inner core, and many other parameters that are still not tightly constrained, and, thus, would introduce large uncertainties, likely comparable to the values themselves. We have added this consideration to the text (lines 258-262). Yet, we still can propose a trend for variation of their concentrations within the inner core. The increment is related to the shape of the solid solution line between the solidus and the liquidus.

Question:- Two recent papers discuss the effects of carbon and silicon incorporation in hcp Fe and the melting properties of the system respectively (Pamato et al. 2020: Equation of State of hcp Fe-C-Si Alloys and the Effect of C Incorporation Mechanism on the Density of hcp Fe Alloys at 300 K and Hasegawa et al. 2020 Liquidus Phase Relations and Solid-Liquid Partitioning in the Fe-Si-C System Under Core Pressures). A comparison of the equations and density used, along with the partitioning data used for the concentration calculations would have been appropriate, can the authors provide one? And I imagine a comparison with Pamato et al. might have been in the work cited in reference 5, but because that work seems to be not accessible it's hard to explain why the parameters of the 300K EoS from that study are different from those reported in the current manuscript.

Answer: The detailed comparison of the EOS parameters with the results of Pamato et al. (2020) would be indeed interesting, but it is beyond the scope of the present study. EOS parameters, as well as density, are not involved in the EVPSC modeling, and hence, anisotropy calculation. Viscosity calculation indeed includes density, though that is merely one parameter among several others. However, as previously mentioned, we tried to put the density of the

alloy from Pamato et al. (2020) to the viscosity calculation and the obtained value of viscosity within the uncertainty range.

We have not performed concentration calculations based on partitioning as we mentioned previously, and added the reason to the text (lines 258-262). Such concentration calculation would require many parameters that are not currently available. Therefore, instead of calculation, we employed a qualitative trend based on general considerations of the solidus curve.

Response to Reviewer 2:

This manuscript shows X-ray diffraction measurements on an Fe-Si-C alloy at high pressure and high temperature. The sample's deformation behavior is extrapolated to Earth's core conditions. The main conclusion is that the properties of the specific composition analyzed match the seismic anisotropy in the Earth's core. The effort the authors devoted to all the steps of the experiment and the analysis is very clear. The careful sample preparation in piston-cylinder apparatus followed by microprobe compositional analysis is very much appreciated, as well as the difficult resistive heating experiments with a panoramic diamond anvil cell. The radial X-ray diffraction data analysis is also challenging and the authors clearly show that they master all these complicated steps.

We thank Reviewer #2 for the positive evaluation of our experimental work.

However, the results are shown quickly, leaving the reader with a long list of unaddressed questions. A solid interpretation and understanding of the results is missing, while much effort is devoted to discussing extrapolated parameters in relation to Earth's core. Unfortunately, the extrapolations are presented without confidence intervals, leaving the reader uncertain about their true significance. The structure of the manuscript is also quite unfriendly: Conclusions are missing, in the Methods are presented results (for example lines 332 – 338) and the reader continuously has to jump between methods, supplementary material, figures captions and literature to have basic fundamental information of what is being presented. I thus believe that the authors need to revisit their work strengthening the robustness of their findings as well as improving the form of the manuscript before it can be published.

Answer:

Following the reviewer's suggestions we have reorganized the structure of the manuscript to increase its clarity and added a conclusion (lines 266-272) Added/modified parts are highlighted in yellow, and rearranged parts are highlighted in turquoise. The responses to specific points of criticism are provided below.

Major comments: -

Line 99:

Question: The authors state that the lattice strain parameter at 300 K and at 1100 K overlap within the uncertainty up to 60 GPa. In Line 110 the authors determine the variation of the lattice strain parameter as a function of temperature at 36 GPa. These two findings are in contradiction. From Fig. 2 the dQ/dT value seem to vary a lot with pressure (as the authors state, the 1100 K values decrease with pressure while the 300 K values increase). What is the significance of a dQ/dT calculated only at one pressure? How does this compare with the other pressures (even if with only two measured temperatures)? Moreover, the dQ/dT is compared with the literature, but the PT conditions of validity of the literature value are not mentioned.

Answer: There was indeed an unclear statement regarding the temperature effect on the lattice strain parameter. We observe an overlap of $\langle Q(hkl) \rangle$ values at 300 K and 1100 K within the error bars if we use lattice strains of (100), (002), and (103) lattice planes for its determination. We plot the $\langle Q(hkl) \rangle$ averaged only over these planes because we can use exclusively their reflections for the entire pressure range at 300 K. Therefore, at 1100 K in Fig. 1 (former Fig. 2) we also show $\langle Q(hkl) \rangle$ calculated only from (100), (002), and (103) lattice strains for a

correct comparison. We have modified the text accordingly to clarify this point (lines 77-81, 117-118). For the calculation of $\frac{\partial Q}{\partial T}$ we were able to use a more complete set of planes, namely (100), (002), (101), (102), (110), (103) and (112).

The decrease of $\langle Q(hkl) \rangle$ at 1100 K above 60 GPa occurs because of the precipitation of Fe_3C , which means the release of carbon from the alloy and we find this data not representative for the calculation of $\frac{\partial Q}{\partial T}$.

The correct comparison of $\frac{\partial Q}{\partial T}$ at different pressures is difficult with the available data since at 300 K we have only a limited number of reflections, which is significantly smaller than that at 1100 K.

We agree that the determination of $\frac{\partial Q}{\partial T}$ at different pressures would be interesting, however, even its determination along one isobar is already quite challenging. Please note that in most of the previous studies $\frac{\partial Q}{\partial T}$ was simply assumed to be zero. We have used 7 temperature points, that were distributed relatively uniformly in the range of 373-1073 K to calculate $\frac{\partial Q}{\partial T}$ at 36 ± 3 GPa (**Supplementary Fig. 2**). To the best of our knowledge, there is only one determination of $\frac{\partial Q}{\partial T}$ by Brennan et al. (2021) at 43-46 GPa and 300-1800 K. However, their results are effectively divided into two clusters, with one point at 300 K and others at 1400-1800 K (Brennan et al. (2021), Figure S1) with a 1000 K gap. Moreover, Brennan et al. (2021) obtained their results by laser heating which is usually associated with large temperature gradients, while we used uniform resistive heating. Following the reviewer's suggestion, we have added the pressure-temperature conditions of Brennan et al. (2021) to the main text (lines 101-102).

Line 101:

Question: The decrease is very small and within the uncertainty. Is the same decrease visible for the considered lattice planes? Or is it just an effect of the average? The authors claim this decrease is due to the formation of cementite. Can they quantify from the diffraction patterns the amount of cementite present in their sample and check if this is consistent with their hypothesis? If cementite is forming it means that the alloy is heterogeneously depleted in carbon and more silicon rich. How would this affect the results shown?

Answer: We see either a decrease or the same level of individual $Q(hkl)$ at 1100 K and above 60 GPa (please see **Supplementary Table 2**). The decrease of $\langle Q(hkl) \rangle$ at 1100 K in **Fig. 1** is indeed small, yet without precipitation of Fe_3C , it is not expected to occur at all. Quantification of the amount of cementite (Fe_3C) unfortunately is not possible since cementite crystallizes in a small number of single-crystal grains (see **Supplementary Fig. 1a**) that precluded a proper Rietveld refinement. We observe a qualitative increase in the amount of Fe_3C diffraction lines with time and, therefore, with pressure increase. To minimize the influence of the compositional changes of the sample on our results, we do not use data above 60 GPa for comparison with the EVPSC modeling, so it does not affect the results.

Question: Fig.2 : Fe-5Si data from ref [18] are shown, while in Supplementary Fig. 3 are shown Fe-5Si and Fe-9Si data from ref [10]. If they are the same data, please refer to them consistently.

Answer: The reference numbers in the main text and supplementary of the Nature Communication journal are independent. Following the reviewer's suggestion to improve clarity, we modified the reference in **Supplementary Fig. 5** (former **Supplementary Fig. 3**) to Brennan et al. (2021).

Question: Why are Fe-9Si data missing in Fig.2? Please specify in the caption the temperature of all data sets.

Answer: We have not displayed Fe-9Si initially so as not to overload the plot, since Fe-9Si data points are similar to Fe-5Si. Now we have modified **Fig. 1** (former **Fig. 2**) and added Fe-9Si. We have added the temperature conditions to the caption (lines 119, 122).

Question: Supplementary Fig.1: it would be useful to show here which lattice planes were used for the determination of Q.

Answer: We have marked the Laue indices of the planes that were used for the calculation of $Q(hkl)$ in **Supplementary Fig. 1**.

Minor comments: -

Line 65:

Question: probably here it's worth mentioning that the sample is polycrystalline and synthesized at high pressure and temperature. Why did the authors choose this specific composition? –

Answer: We have modified the text accordingly (lines 65-66), mentioning that the sample is polycrystalline and it was obtained at high-temperature high-pressure conditions. We have chosen this composition because of its proximity to the suggested compositions for the Earth's inner core (e.g., Li et al. (2018), Miozzi et al. (2020)) (lines 66-67).

Line 78:

Question: The scale indicating the multiples of mrd in Fig. 1 goes maximum to 4. I understand that the scale is not the same for the 300 K and the 1100 K cases. –

Answer: We have modified **Fig. 2** (former **Fig. 1**) and now there are two different scales for 300 K and 1100 K.

Line 137:

Question: Too many information missing. I suggest you to change the sentence with something like: "Finally, by combining the stiffness tensor components from the literature and the extrapolated value of Q, we calculated the yield strength of Fe-2Si-0.4C at core conditions."

Answer: We have modified the text accordingly (lines 177-179).

Question: The uncertainty of this value, as shown in Supplementary Fig. 4 looks very small. Have the authors propagated the uncertainties over the extrapolation? In any case, it is worth mentioning how this uncertainty was determined, before claiming that it is smaller than the previous one.

Answer: The uncertainty of the yield strength extrapolations comes from the uncertainty of $\langle Q(hkl) \rangle$ at high-pressure high-temperature conditions and the uncertainties of the shear

modulus value that originate from the uncertainties of the stiffness tensor components. Although the EVPSC modelling does not directly provide the uncertainties for the $\langle Q(hkl) \rangle$ values, in order to estimate the uncertainties of the model, we have varied all the stiffness tensor components within their uncertainties (**Supplementary Table 13**). That provides us the ability to calculate the range of possible $\langle Q(hkl) \rangle$ values up to the core conditions. We have added the propagations of uncertainties to the determination of $\langle Q(hkl) \rangle$ at high pressure in **Supplementary Fig. 5**. Also we have added the uncertainty of the determination of $\frac{\partial Q}{\partial T}$ (**Supplementary Fig. 2**) Applying the uncertainties of $\langle Q(hkl) \rangle$ modeling at high pressure, we changed the uncertainties propagation of the yield strength (**Fig. 5**). We have modified the text accordingly (lines 179-182).

- Line 222:

Question: some “shades” are visible in the backscattered electron image, while the compositions look completely homogeneous. Are the two images taken on the exact same spot? Can you add a compositional scale to the elemental X-ray distribution maps? What do the colors represent?

Answer: The compositional maps were taken from the same spot as the backscattered electron image. We have taken compositional maps only in the squared form (the centers of images are the same). The differences in the SEM image in lighter and darker areas are caused by electron channelling due to different crystal orientations. Interpretation of the color bar is: red pixels have a high quantity of the element while blue pixels have a low quantity of the element. We have added these clarifications to the caption of **Supplementary Fig. 6**. Please note, that the color maps there are qualitative and, thus, the quantitative legend cannot be provided.

REVIEWER COMMENTS

Reviewer #1 (Remarks to the Author):

I have now looked at the revised version of the manuscript “Depth-dependent anisotropy in the Earth’s inner core linked to chemical stratification” by Kolesnikov and co-authors. Before getting into the details, I want to apologize with the authors for the delay of my review and thank them for the thorough response to my previous comments.

The authors made a great effort in making the manuscript more accessible and clearer for the reader and I think it has improved in respect to the first version, but it still requires some work. I have few more questions and some comments that I think will improve the paper. After this I think the work will be a nice contribution to the field.

We thank Reviewer #1 for the positive evaluation of our revised work. We have further refined the structure of the manuscript: added or modified parts are highlighted in yellow, and moved parts are highlighted in turquoise.

The authors start discussing the results without mentioning what is the stable crystallographic structure and potentially the chemistry of the sample. Only after a while hcp Fe-2Si-0.4C starts being mentioned. It would be great if a mention to the structure and chemistry could be added so that a reader can understand better.

We have unified the sample naming throughout the text to hcp-Fe-2Si-0.4C, where applicable.

On the iron side, starting at 36 GPa with the high temperature data the authors might cross the fcc stability field before the hexagonal structure becomes the stable one? It would be great if the author can comment on this. The only mention of the chemistry comes in line 89 where the manuscript states “We observe the appearance of new reflections in the X-ray diffraction patterns that correspond to the formation of Fe₃C” and then in line 90 there is a reference about hcp being stable with Fe₃C but there are no reference to what are the structures are before that or even in the image of the diffraction patterns in the supplementary.

We have clarified the transformation path for the samples in the lines 75-77, and added example XRD patterns in **Supplementary Fig. 1**. The starting material was bcc-structured Fe-2Si-0.4C, which began to transform into the hcp phase above 10 GPa and 300 K. The process of transformation completes at ~27 GPa.

During the resistive heating experiments, we indeed observed face-centered cubic (fcc) structure; however, it was as a less abundant component, compared to hcp phase, and completely disappears at ~55 GPa and 1100 K. We added the reference to lines 77-80.

It should also be mentioned what is the initial structure of the alloy (maybe in the methods where the sample is described?) and not only in the caption of supplementary Figure 3. Additionally, the Fe-Si-C alloy is defined in different ways across the paper, e.g., it starts as hcp Fe-2Si-0.4C, then in line 360 it is defined as hcp Fe alloy then in line 190 and further down it is referred to as Fe-2Si-0.4C. I'd suggest using one of the three across all the parts of the paper to improve clarity for the reader.

The initial structure of the sample (bcc) is now explicitly mentioned in line 75 and in the Methods section (lines 354-355), in addition to the **Supplementary Fig. 3** caption. We have also unified the sample naming as hcp-Fe-2Si-0.4C throughout the text, where applicable.

On this note, I think Figure Supplementary 1 needs some clarity. This is the first image in which raw data are shown, it would be great if there can be some more details to make it more understandable. What is the color scale? And what are the other phases for the 1100 K data? Why in the figure at 1100 K only the peaks of Fe₃C are shown? I would also add to the caption of figure b that it is to show the contribution of the gasket to the diffraction pattern, as readers might wonder why there is no sample listed.

We have moved former **Supplementary Fig. 1** to the main text as **Fig. 1** and added additional legends to improve clarity. In the diffraction pattern collected at 300 K, the identified phases are Be, BeO, and the hcp-Fe-2Si-0.4C sample (marked by Laue indices at the top). At 1100 K, the observed phases are Au (pressure standard), Au (outside of the pressure chamber), Fe₃C impurity, and the hcp-Fe-2Si-0.4C sample (also marked by Laue indices at the top). Previously, these phases were indicated by symbols, described in the figure caption; we have now added their names alongside the symbols for clarity. The color scale represents the square root of intensity, as stated in the legend.

“Carbon-free compositions, like pure Fe₁₇, Fe-Ni alloys₂₄ and Fe-5 wt % Si₂₅, display smaller $\langle(hkl)\rangle$ values compared to Fe-2Si-0.4C (Fig. 1a). Hence, the partial loss of carbon from the starting composition during heating likely causes the decrease in the averaged lattice strain parameter in the iron-silicon-carbon alloy.”
The data the authors are comparing with, are data collected with another scope in mind and consequently trying to be as hydrostatic as possible and with another beamline geometry (axial and not radial). Because their conclusion for this portion is based on the comparison, can the authors comment on how reliable it is to compare datasets collected in such different ways?

We consider the comparison reliable, because in all cited studies, namely pure Fe (Gleason & Mao (2013)), Fe-Ni alloys (Reagan et al. (2018)) and Fe-5 wt % Si (Brennan et al. (2021)), $\langle Q(hkl)\rangle$ values were measured under intentionally non-hydrostatic conditions using radial X-ray diffraction geometry. Our experiments were conducted under the same type of conditions, ensuring that the comparison is meaningful.

In line 252 the authors mention the concentration of C in the solid solution changing with temperature

The data proposed were collected in axial standard geometry, does it matter?

The discussion in line 302 (former line 252) concerns the Fe-C phase relations at inner core conditions. The results of Mashino et al. (2019) obtained using axial geometry under quasi-hydrostatic conditions are directly applicable because the Earth's inner core is also under quasi-hydrostatic conditions, with the hydrostatic pressure exceeding $3.29 \cdot 10^{11}$ Pa and estimated shear stresses $<10^6$ Pa, i.e., at least five orders of magnitude lower. Therefore, the change of C concentration with temperature inferred from Mashino et al. (2019) remains relevant for inner core conditions.

Line 46: “explain the density deficit compared to reported by seismic models” Seems like maybe only compared to or reported by should be kept?

We have modified the sentence accordingly (line 46).

Line 76 and 78: $Q(hkl)$ are written with different formalisms.

Here, in lines 81 and 83 (former 76, 78) we use different formalisms in order to make a distinction between the lattice strain parameter $Q(hkl)$ of individual lattice planes and the averaged lattice strain parameter $\langle Q(hkl) \rangle$ that is averaged from $Q(hkl)$ of several lattice planes.

Line 81: Typo: “lattice stains”

We appreciate the thorough reading of Reviewer#1 and have corrected the typo.

Line 87: “Compared to presented” there is a subject missing?

We have modified the statement in line 103 (former line 87) to make it clearer.

Line 90: “co-existence of Fe₃C and hcp-Fe-Si-C alloy” I think Miozzi et al. observed hexagonal structured iron from the diffraction pattern and they used the volume increase to infer the presence of Si and C in the structure of the iron. Is the same structuring and volume observed for the “material” in this work to confirm that indeed is a similar phase?

We observe the same structure (i.e., hcp) and similar unit cell volumes as Pamato et al. (2020) and Miozzi et al. (2020), as we show in **Supplementary Fig. 6**.

Line 97: “providing only the upper boundary for $\langle(hkl)\rangle$.” What do the authors mean?

Here in line 114 (former line 97), we mean that since $\frac{\partial Q}{\partial T}$ is a negative value, therefore, previous assessments that considered $\frac{\partial Q}{\partial T}$ as simply zero, provided only a maximum value, i.e., an upper bound for $\langle Q(hkl)\rangle$.

Supplementary figure 1: The spacing between top and bottom needs to be bigger. Both the figures look like it is only one and not two.

We have modified **Fig. 1** (former **Supplementary Fig. 1**) accordingly.

Line 176: “to derive $\langle(hkl)\rangle$ values that are further utilized the constrained $\frac{\partial Q}{\partial T}$ to calculate $\langle Q(hkl)\rangle$ at core temperatures.” Not clear what the authors meant here, maybe it was “to constrain $\frac{\partial Q}{\partial T}$ ”?

We first use our EVPSC deformation model to obtain $\langle Q(hkl)\rangle$ at core pressures, which are higher than the experimentally observed range. We then apply the previously determined $\frac{\partial Q}{\partial T}$ to these values to account for the effect of temperature, thereby estimating the averaged lattice strain parameter of hcp-Fe-2Si-0.4C at core pressures and temperatures. We have modified the sentence for clarity (lines 225-227).

Lines 193-196: “Within the pressure range investigated in our study, however, hcp-Fe-2Si-0.4C displays higher strength than the iron-silicon alloys studied by Brennan et al. (2021)²⁵ (Fig. 1b). This discrepancy likely arises from the extrapolation of $\langle(hkl)\rangle$ values to core conditions.”. I am confused by this sentence, how can a discrepancy in the studied pressure range depend on an extrapolation at core condition? Maybe the authors meant something else?

Within the experimentally observed pressure range (**Fig. 2b**) hcp-Fe-2Si-0.4C demonstrates higher strength than the Fe-Si alloys studied by Brennan et al. (2021). However, when extrapolated to core conditions (**Fig. 6**), the extrapolated strength of the Fe-Si alloys is much higher than that of hcp-Fe-2Si-0.4C. We attribute this apparent discrepancy to the linear extrapolation applied by Brennan et al. (2021). We have modified the sentence for clarity (lines 245-246).

Line 232: “experimental V_p anisotropy”. If the V_p is estimated (line 231) from the texture modelling I think it shouldn't be called “experimental”. It might lead people to think it was an experimentally observed quantity, while it is an estimate, from a model in turn based on the experiments.

We agree with the reviewer that this may be confusing and therefore we have modified the sentence to clarify that the velocities are derived from the experimental textures (line 283, former line 232). Anisotropy at core conditions obtained using the EVPSC texture modeling are referred to as extrapolated like in line 281 (former line 231) and line 287.

Reviewer #2

(Remarks to the Author):

The authors indeed put some effort in trying to ameliorate the structure of their manuscript, which now reads better than the previous version. However, the manuscript still does not stand alone as too many essential information (especially Figures) have to be looked for in the supplementary materials to be able to understand what the authors are talking about.

We thank Reviewer#2 for the positive evaluation of our efforts. We have moved several figures, a table and Discussion from the Supplementary to the main text to improve the presentation of our study. We have further refined the structure of the manuscript: added or modified parts are highlighted in yellow, and moved parts are highlighted in turquoise.

Moreover, the authors tried to answer the question about previous line 101 (current 84) referencing to the data in table 2 of supplementary materials. I plotted these data and I performed a linear fit of the average over all the lattice planes indicated in line 86. The result I obtain is $y = 9e-06 x + 0.0053$, which is increasing over pressure and not decreasing as stated in lines 85-87. The statement in line 91 and the related conclusions are thus not supported by the data presented in the paper. The statement in line 91 is also in contradiction with what is shown in Supplementary Figure 5.

I thus feel that still some work has to be done to make this manuscript ready for publication and that Nature Communications is probably not the suitable choice of journal.

We appreciate Reviewer#2's effort to verify the reproducibility of our results. We performed the same procedure, plotting the averaged lattice strain parameter $\langle Q(hkl) \rangle$, calculated from planes (100), (002), (101), (102), (110) and (103) in **Response Figure 1** using the data reported in **Supplementary Table 2**. For convenience, we reproduce below **Supplementary Table 2**.

We observe that $\langle Q(hkl) \rangle$ increases up to ~60 GPa and then decreases, as noted in lines 101-104 (former lines 84-87), 107-108 (former line 91). A linear fit of $\langle Q(hkl) \rangle$ versus pressure over the entire dataset yields a negative slope of $-6(\pm 2) \cdot 10^{-6} \text{ GPa}^{-1}$. When considering only the decreasing region above 60 GPa, a linear fit gives a slope of $-1.4(\pm 0.3) \cdot 10^{-5} \text{ GPa}^{-1}$ (**Response Figure 1**). These results are fully consistent with our statements in lines 101-104 and 107-108 of the main text (former lines 84-87 and 91).

We were unable to reproduce the positive slope ($y = 9e-06 x + 0.0053$) reported by Reviewer#2. To improve clarity, we have modified lines 107-108 (former line 91) to explicitly highlight that the decrease of $\langle Q(hkl) \rangle$ at 1100 K occurs *above 60 GPa*, eliminating the apparent contradiction with **Fig. 4** (former **Supplementary Fig. 5**).

Response Figure 1. Pressure evolution of the averaged lattice strain parameter $\langle Q(hkl) \rangle$ obtained from planes (100), (002), (101), (102), (110) and (103) at 1100 K (**Supplementary Table 2**). The red line is the linear fit of all the data points, with the fitting equation $\langle Q(hkl) \rangle(P) = -6(\pm 2) \cdot 10^{-6} \cdot P + 0.0061(\pm 0.0002)$. The blue line represents the linear fit of $\langle Q(hkl) \rangle$ at pressures above 60 GPa, where an actual decrease is observed, with a fitting equation $\langle Q(hkl) \rangle(P) = -1.4(\pm 0.3) \cdot 10^{-5} \cdot P + 0.0067(\pm 0.0003)$.

Supplementary Table 2. Pressure evolution of the lattice strain parameters $Q(hkl)$ and averaged $\langle Q(hkl) \rangle$ of hcp-Fe-2Si-0.4C at 1100 K.

P, GPa	ΔP , GPa	$Q(100)$	$\Delta Q(100)$	$Q(002)$	$\Delta Q(002)$	$Q(101)$	$\Delta Q(101)$	$Q(102)$	$\Delta Q(102)$	$Q(110)$	$\Delta Q(110)$	$Q(103)$	$\Delta Q(103)$	$\langle Q(hkl) \rangle$	$\Delta \langle Q(hkl) \rangle$
36.33	0.05	0.0073	0.0003	0.0082	0.0001	0.0076	0.0001	0.0018	0.0003	0.0031	0.0004	0.0053	0.0003	0.0056	0.0011
37.28	0.05	0.008	0.0003	0.0078	0.0002	0.0061	0.0001	0.0013	0.0004	0.003	0.0003	0.0055	0.0002	0.0053	0.0011
38.9	0.04	0.007	0.0005	0.0078	0.0002	0.0063	0.0001	0.0015	0.0002	0.0047	0.0003	0.0047	0.0002	0.0053	0.0009
40.32	0.05	0.0078	0.0003	0.0078	0.0002	0.006	0.0001	0.0016	0.0001	0.0042	0.0004	0.005	0.0002	0.0054	0.0010
43.2	0.1	0.0085	0.0003	0.0081	0.0002	0.0062	0.0001	0.0024	0.0002	0.0058	0.0003	0.005	0.0001	0.0060	0.0009
44.81	0.07	0.008	0.0003	0.0085	0.0002	0.0061	0.0001	0.0034	0.0002	0.0058	0.0002	0.0043	0.0001	0.0060	0.0008
47.4	0.07	0.0072	0.0004	0.0086	0.0001	0.0055	0.0001	0.0046	0.0003	0.006	0.0003	0.0052	0.0003	0.0062	0.0006
47.9	0.12	0.0068	0.0004	0.008	0.0001	0.0057	0.0001	0.0036	0.0002	0.0057	0.0002	0.0044	0.0001	0.0057	0.0007
49.02	0.07	0.0058	0.0003	0.0082	0.0001	0.0058	0.0001	0.0037	0.0002	0.0064	0.0002	0.004	0.0002	0.0057	0.0007
52.8	0.1	0.0057	0.0002	0.0097	0.0003	0.0054	0.0001	0.0037	0.0006	0.0068	0.0001	0.0041	0.0001	0.0059	0.0009
55.24	0.08	0.0064	0.0003	0.0094	0.0002	0.0054	0.0001	0.0033	0.0004	0.0072	0.0003	0.0042	0.0002	0.0060	0.0009
60.14	0.02	0.0066	0.0003	0.0092	0.0002	0.0052	0.0001	0.004	0.0003	0.008	0.0002	0.004	0.0002	0.0062	0.0009
66	0.08	0.0067	0.0003	0.0086	0.0002	0.0048	0.0001	0.0046	0.0003	0.007	0.0003	0.004	0.0002	0.0060	0.0007
67.5	0.1	0.0056	0.0005	0.0085	0.0002	0.0048	0.0001	0.0044	0.0004	0.0071	0.0002	0.0042	0.0002	0.0058	0.0007
69.27	0.08	0.0057	0.0003	0.0085	0.0001	0.0053	0.0001	0.0038	0.0004	0.0076	0.0002	0.0037	0.0002	0.0058	0.0008
71.1	0.1	0.0057	0.0003	0.0083	0.0001	0.0052	0.0001	0.0041	0.0002	0.0074	0.0002	0.0033	0.0002	0.0057	0.0008
73.86	0.08	0.0046	0.0004	0.0078	0.0001	0.0051	0.0001	0.0043	0.0003	0.0068	0.0005	0.0051	0.0001	0.0056	0.0006
76.46	0.01	0.0047	0.0004	0.0081	0.0001	0.0052	0.0001	0.0051	0.0002	0.007	0.001	0.004	0.0003	0.0057	0.0007
77.78	0.01	0.0053	0.0003	0.0081	0.0001	0.0048	0.0001	0.0048	0.0003	0.0077	0.0005	0.0048	0.0005	0.0059	0.0006
78.6	0.1	0.0053	0.0004	0.0078	0.0001	0.005	0.0001	0.0047	0.0003	0.0064	0.0005	0.0046	0.0004	0.0056	0.0005
81.33	0.01	0.0052	0.0002	0.0074	0.0001	0.0045	0.0001	0.0045	0.0003	0.0071	0.0004	0.0042	0.0006	0.0055	0.0006
83.51	0	0.0062	0.0002	0.0075	0.0001	0.005	0.0001	0.0044	0.0003	0.007	0.0002	0.0043	0.0004	0.0057	0.0006
85.56	0.01	0.0057	0.0003	0.0074	0.0001	0.005	0.0001	0.0045	0.0003	0.0063	0.0002	0.0044	0.0003	0.0056	0.0005
88.6	0.17	0.0061	0.0002	0.0072	0.0001	0.0047	0.0001	0.0041	0.0002	0.007	0.0002	0.0031	0.0002	0.0054	0.0007
90.63	0.11	0.0055	0.0002	0.0071	0.0001	0.0047	0.0001	0.0043	0.0003	0.0078	0.0003	0.0043	0.0003	0.0056	0.0006
92.64	0.13	0.0053	0.0002	0.0071	0.0001	0.0046	0.0001	0.004	0.0002	0.0067	0.0001	0.0034	0.0001	0.0052	0.0006
95.11	0.16	0.0064	0.0001	0.0073	0.0001	0.005	0.0002	0.0042	0.0003	0.0072	0.0002	0.0035	0.0002	0.0056	0.0007
98.1	0.02	0.0055	0.0002	0.0074	0.0001	0.005	0.0002	0.004	0.0003	0.0066	0.0002	0.0045	0.0002	0.0055	0.0005
100.14	0.13	0.006	0.0001	0.0072	0.0001	0.0047	0.0001	0.004	0.0002	0.0067	0.0002	0.0036	0.0001	0.0054	0.0006

Response to reviewers

REVIEWER COMMENTS

Reviewer #1

(Remarks to the Author): I have now looked at the revised version of the paper. I think the manuscript reads well in its current form and I thank the authors for their work. They clarified the points that I had questions on, and made the manuscript approachable despite the short format. I have no further comments.

We thank Reviewer #1 for the positive evaluation of our revised work.

Reviewer #2 (Remarks to the Author):

I apologize with the authors for my mistake trying to verify their results. Their statement about the pressure evolution of the averaged lattice strain parameter $\langle Q(hkl) \rangle$ was indeed correct. The manuscript indeed reads much better now than in the previous versions, and the findings are more clearly presented and discussed.

We thank Reviewer #2 for the positive evaluation of our revised work.

Still, I don't understand the relevance of showing lattice strain parameter averaged on a different set of reflections (Fig. 2 and Fig. 4.), especially given that the trend of the average over pressure changes according to the number of reflections considered: for example, in Fig 2 (average over 3 reflections) the general trend is that $d\langle Q \rangle/dP$ is negative, while in Figure 4 (average over 6 reflections) it is positive.

Fig. 2 shows the comparison of the entire dataset of the derived $\langle Q(hkl) \rangle$ values at 300 K and at 1100 K. Since only 3 lattice planes are available at 300 K, we also used only these 3 lattice planes at 1100 K for a correct comparison. **Fig. 4**, in turn, displays only the data that were used for the comparison with the EVPSC modeling, explicitly excluding the data above 60 GPa that show a negative trend as unreliable. For a more accurate comparison with the EVPSC modeling, we use the full set of available lattice planes. We, thus, believe that both figures are necessary for the correct representation of our data and for the correct use of our data in future studies.

Moreover, I think it would be appreciable to have a more developed and dedicated conclusion paragraph. In conclusion, in this manuscript it is shown how the anisotropy is dependent on the chemistry. Measurements up to about 100 GPa and 1100 K are used to model the deformation mechanisms, yield strength and viscosity of hcp Fe-2Si-0.4C at core conditions. My only reservation is whether such a technical work meets the criteria for a high-impact journal.

We have modified the conclusions accordingly (lines 248-260). We kindly disagree with the reviewer that our work is too technical for a high-impact journal. We modeled the deformation of hcp-Fe-Si-C alloy up to the core pressures, and the results of our modeling qualitatively correspond to the observations at extreme conditions in refs ^{1,2}. We note that our work constraints for the first time the combined effect of two key candidate light elements, C and Si, on the anisotropy of hcp-Fe at high-temperature conditions. Hence, the results allow for a more reliable extrapolation of the anisotropy estimates at inner core conditions compared to previous models based on linear extrapolations and/or room temperature data. Moreover, our results together with literature data for hcp-Fe provide for the first time qualitative constraints on the effect of chemical stratification on the anisotropy pattern of the inner core. We thus consider that these results are of great interest for the broad scientific community.

Response references

1. Ikuta, D. *et al.* Sound velocity of hexagonal close-packed iron to the Earth's inner core pressure. *Nat Commun* **13**, 7211 (2022).
2. Sakamaki, T. *et al.* Constraints on Earth's inner core composition inferred from measurements of the sound velocity of hcp-iron in extreme conditions. *Sci. Adv.* **2**, e1500802 (2016).